# Simplicial Embeddings Improve Sample Efficiency in Actor–Critic Agents

**Johan Obando-Ceron**[*1,2]    **Walter Mayor**[*3]    **Samuel Lavoie**[1,2]    **Scott Fujimoto**[1,4]
**Aaron Courville**[1,2,5]    **Pablo Samuel Castro**[1,2]

[*]Equal contribution    [1]Mila – Québec AI Institute    [2]Université de Montréal
[3]Independent Researcher    [4]McGill University    [5]CIFAR AI Chair

{jobando0730, waltermayor, samuel.lavoie.m}@gmail.com
{courvila, pablo-samuel.castro}@mila.quebec
scott.fujimoto@mail.mcgill.ca

## Abstract

Recent works have proposed accelerating the wall-clock training time of actor-critic methods via the use of large-scale environment parallelization; unfortunately, these can sometimes still require large number of environment interactions to achieve a desired level of performance. Noting that well-structured representations can improve the generalization and sample efficiency of deep reinforcement learning (RL) agents, we propose the use of *simplicial embeddings*: lightweight representation layers that constrain embeddings to simplicial structures. This geometric inductive bias results in sparse and discrete features that stabilize critic bootstrapping and strengthen policy gradients. When applied to FastTD3, Fast-SAC, and PPO, simplicial embeddings consistently improve sample efficiency and final performance across a variety of continuous- and discrete-control environments, without any loss in runtime speed. **Source code here.**

*"Order is not imposed from the outside, but emerges from within[1]."*

*— Ilya Prigogine*

## 1 Introduction

Deep reinforcement learning (deep RL) has delivered impressive progress in continuous control, enabling agile locomotion (Smith et al., 2022; Zhuang et al., 2023; Margolis et al., 2024) and dexterous manipulation (Popov et al., 2017; Akkaya et al., 2019; Luo et al., 2025). Yet a persistent tension remains between *training speed* (wall-clock efficiency) and *sample efficiency* (the number of environment interactions). Some modern agents such as TD-MPC2 (Hansen et al., 2024) and SR-SPR/BBF (D'Oro et al., 2022; Schwarzer et al., 2023) achieve strong returns with relatively few interactions, but demand substantial compute and engineering complexity. In contrast, recent fast actor–critic variants have scaled throughput with massive parallelization (Li et al., 2023; Singla et al., 2024; Gallici et al., 2025; Seo et al., 2025). While methods such as FastTD3 (Seo et al., 2025) rapidly solve humanoid benchmarks, they require far more interactions to reach comparable performance. Similar limitations have been observed in Parallel Q-Learning (Li et al., 2023) and large-scale actor–critic frameworks such as IMPALA and SEED RL (Espeholt et al., 2018; 2020). This trade-off limits applicability in domains where interactions are expensive and time is constrained, such as robotics.

A natural objection is that, in modern simulators, environment steps are cheap and can be generated in massive parallel batches, so sample efficiency is less important. However, this view overlooks

---

[1]This perspective resonates with deep RL: stability cannot be forced solely through more compute, heavier regularizers, or larger critics. Instead, inductive biases that shape the geometry of representations can allow order to *emerge from within*, leading to more stable critics and more efficient policies under non-stationarity.

several practical and scientific concerns. First, algorithms that are data-hungry in simulation rarely transfer well to real-world scenarios (Tobin et al., 2017; Akkaya et al., 2019). Second, large-scale parallelization requires substantial compute and energy resources, raising both efficiency and sustainability concerns (Schwartz et al., 2020; Henderson et al., 2020). Third, sample efficiency is closely tied to generalization: agents that exploit structure from fewer trajectories tend to be more robust under distributional shifts (Zhang et al., 2018; Yao et al., 2025). Moreover, in high-dimensional simulators such as Isaac Gym, each step can be significantly more expensive, compounding inefficiency as tasks grow harder (Makoviychuk et al., 2021; Rudin et al., 2021). These issues highlight why sample efficiency remains central even in the era of massively parallel deep RL.

Shaping representations with auxiliary losses (Anand et al., 2019; Laskin et al., 2020; Schwarzer et al., 2021; Castro et al., 2021; Fujimoto et al., 2023) has been shown to improve sample efficiency in deep RL. However, such methods increase algorithmic complexity and add computational overhead through extra forward and backward passes (Fujimoto et al., 2023). Alternatively, architectural components, such as convolutions (Fukushima, 1980; LeCun et al., 1989) and attention (Bahdanau et al., 2016), can be used to induce structure leading to desirable downstream properties.

Discrete and sparse representations have several desirable properties in comparison to their dense and continuous counterparts. Notably, sparse and discrete representations increase robustness to noise (Donoho et al., 2006), training stability by reducing catastrophic interference (Liu et al., 2019), sample efficiency (Fumero et al., 2023), interpretability (Murphy et al., 2012; Lavoie et al., 2023; Wabartha & Pineau, 2024) and improved generative modeling (Lavoie et al., 2025). In this work, we posit that several of those properties are beneficial in the context of reinforcement learning. Within RL, several successful agents like TD-MPC2, Dreamer V2/V3, IQRL, and MAD-TD (Hansen et al., 2024; Alonso et al., 2024; Hafner et al., 2025; Scannell et al., 2024; Voelcker et al., 2025) use discrete latents; yet they typically rely on auxiliary model-based losses to shape the representation.

While several methods exist for learning discrete representations explicitly (Jang et al., 2017; Maddison et al., 2017; van den Oord et al., 2017), these methods use straight-through estimation (Bengio et al., 2013) which is a biased gradient estimator. Fortunately, discretization may be implicitly induced via Simplicial Embeddings (SEM) (Lavoie et al., 2023), an architectural component that partitions a latent representation into a sequence of $L$ simplices. SEM is fully differentiable, thus avoiding the negative effect of explicit discretization while enacting some of the desirable properties of discrete and sparse representations. Concretely, we show that SEM improves both data efficiency and asymptotic performance across diverse environments such as Isaac Gym (Makoviychuk et al., 2021), HumanoidBench (Sferrazza et al., 2024), and the Arcade Learning Environment (Bellemare et al., 2013), while preserving (and often improving) wall-clock speed.

## 2 PRELIMINARIES

### 2.1 ACTOR–CRITIC REINFORCEMENT LEARNING

We consider a standard Markov decision process (MDP) defined by the tuple $\mathcal{M} = (\mathcal{S}, \mathcal{A}, P, r, \gamma)$, with state space $\mathcal{S}$, action space $\mathcal{A}$, transition distribution $P(s'|s, a)$, reward function $r : \mathcal{S} \times \mathcal{A} \to \mathbb{R}$, and discount factor $\gamma \in [0, 1)$. The objective is to maximize the expected discounted return $J(\pi) = \mathbb{E}_\pi \left[ \sum_{t=0}^\infty \gamma^t r(s_t, a_t) \right]$, where the agent follows a policy $\pi(a|s)$. Actor–critic methods maintain both a parameterized policy $\pi_\theta(a|s)$ (the actor) and an action-value function $Q_\phi(s, a)$ (the critic). The critic is trained to minimize the Bellman error

$$\mathcal{L}_Q(\phi) = \mathbb{E}_{(s,a,r,s')\sim\mathcal{D}}\left[ \left( Q_\phi(s,a) - y \right)^2 \right], \qquad y = r + \gamma \, \mathbb{E}_{a'\sim\pi_\theta(\cdot|s')}\left[ Q_{\phi^-}(s', a') \right], \quad (1)$$

where $\phi^-$ denotes target network parameters and $\mathcal{D}$ is a replay buffer. The actor is updated via the policy gradient defined as $\nabla_\theta J(\pi_\theta) = \mathbb{E}_{s\sim\mathcal{D},a\sim\pi_\theta}\left[ \nabla_\theta \log \pi_\theta(a|s)\, Q_\phi(s,a) \right]$. While this can be effective, bootstrapped training is notoriously fragile. Errors in $Q_\phi$ propagate recursively through the target $y$, and when the representation used to compute $Q_\phi$ is poorly conditioned, these errors amplify and cause divergence or collapse (Fujimoto et al., 2018).

A recent line of work has sought to reduce the *wall-clock* cost of actor–critic training. FastTD3 (Seo et al., 2025) builds on TD3 (Fujimoto et al., 2018) by leveraging (i) parallel simulation across many

environment instances, (ii) large-batch critic updates, and (iii) algorithm design choices like distributional critics (C51) (Bellemare et al., 2017), noise scaling and clipped double Q-learning (CDQ) (Fujimoto et al., 2018). Together, these design choices enable high-throughput training while retaining stable convergence, although FastTD3 (Seo et al., 2025) still remains sample-inefficient (see App. C for more details).

Policies and critics often rely on latent representations extracted from raw states (Lesort et al., 2018). Formally, an encoder $f_\psi : \mathcal{S} \to \mathbb{R}^d$ maps observations $s$ into embeddings $z = f_\psi(s)$, which are then consumed by either the critic, the actor, or both depending on the architecture. Some methods share a common encoder across actor and critic (e.g., SAC (Haarnoja et al., 2018), DrQ (Yarats et al., 2021), DrQ-v2 (Yarats et al., 2022)), while others (e.g., DDPG (Lillicrap et al., 2015), FastTD3 (Seo et al., 2025)) maintain separate encoders. Regardless of parameter sharing, these representations play a central role in learning (Garcin et al., 2025). The critic estimates values $Q_\phi(s, a) \equiv Q_\phi(f_\psi(s), a)$, and the actor conditions its policy $\pi_\theta(a|s) \equiv \pi_\theta(a|f_\psi(s))$ on the chosen embedding. Ideally, $z$ should preserve the Markov property and expose predictive features of the reward $r$ and dynamics $P$.

Yet the choice and stability of such embeddings is far from guaranteed. When unconstrained, learned representations can introduce severe pathologies that destabilize value learning. For example, if $\|f_\psi(s)\| \to \infty$, critics may extrapolate to arbitrarily large Q-values outside the support of the replay buffer, inflating the Bellman error. Formally, if $Q_\phi(z, a) = w^\top z + b$ with linear heads, then $\|Q_\phi\| \to \infty$ as $\|z\| \to \infty$, leading to exploding targets $y$ and divergent gradients. Similarly, if $z$ exhibits strong correlations or degenerate directions, the critic's regression problem becomes ill-conditioned: the covariance matrix $\Sigma = \mathbb{E}[zz^\top]$ may approach singularity, amplifying variance in temporal-difference updates. These phenomena are empirically linked to representation collapse, where value estimates drift irrecoverably and policy updates follow unstable gradients (Moalla et al., 2024; Castanyer et al., 2025).

## 2.2 SIMPLICIAL EMBEDDINGS

Simplicial embeddings (SEM; Lavoie et al., 2023) provide a lightweight inductive bias on representation geometry by constraining latent codes to lie on a product of simplices. Concretely, given encoder outputs $f_\psi(s) \in \mathbb{R}^{L \times V}$, the latent vector is partitioned into $L$ groups of size $V$, and a softmax is applied within each group:

$$\tilde{z}_{\ell,v} = \frac{\exp(z_{\ell,v}/\tau)}{\sum_{v'=1}^{V} \exp(z_{\ell,v'}/\tau)}, \quad \forall \ell \in \{1, \ldots, L\}, \ v \in \{1, \ldots, V\}, \tag{2}$$

where $\tau > 0$ is a temperature parameter controlling the degree of sparsity. The resulting embedding $\tilde{z}$ lies in the product space $\Delta^{V-1} \times \cdots \times \Delta^{V-1}$, i.e., $L$ categorical distributions of dimension $V$. This transformation ensures boundedness through group-wise normalization, induces sparsity as softmax competition (sharpened at low $\tau$) drives near one-hot encodings, and promotes group structure by partitioning features into modular subspaces akin to mixtures-of-experts (Shazeer et al., 2017; Ceron et al., 2024c; Willi et al., 2024). In self-supervised learning and downstream classification, SEM has been shown to stabilize training and improve generalization, particularly in low-label and transfer settings (Lavoie et al., 2023). SEM does not rely on auxiliary losses or reconstruction terms; akin to an activation function, it only modifies the embedding geometry with the group-wise softmax, limiting computational overhead.

## 3 NON-STATIONARITY AMPLIFIES REPRESENTATION COLLAPSE

Several works have shown that non-stationarity can lead to severe degradation of learned representations across different domains (Lyle et al., 2022; Kumar et al., 2021a; Lyle et al., 2025; Castanyer et al., 2025). In supervised learning, label noise and distribution shifts can induce representation collapse, where features lose diversity and neurons become inactive (Li et al., 2022; Sokar et al., 2023; Dohare et al., 2024). Similar observations have been made in deep RL: *the constantly changing data distribution, induced by an evolving policy, exacerbates this phenomenon, often resulting in unstable critics and poor generalization* (Nauman et al., 2024a; Kumar et al., 2021a). These studies suggest that collapse is not an isolated pathology of specific architectures, but a general failure mode that emerges when training signals are non-stationary. In App. B we provide a formal analysis that demonstrates the relationship between non-stationarity and neuron dormancy.

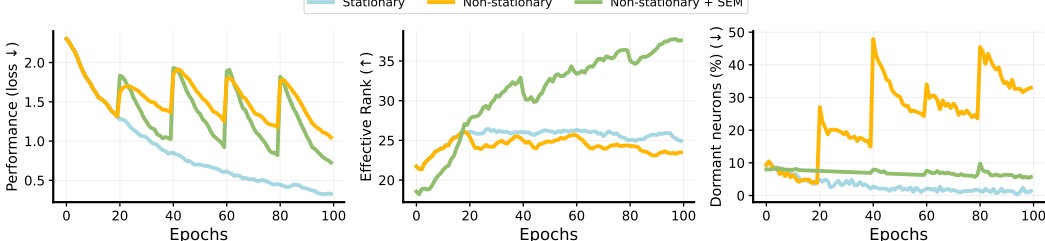

Fig. 1: **Training dynamics on CIFAR-10 with stationary vs. non-stationary targets.** In the stationary regime (fixed targets), losses decrease smoothly, neuron dormancy and effective rank remains controlled, suggesting stable representation learning. In the non-stationary regime (targets shuffled every 20 epochs), the model exhibits higher variance in losses, increased dormant neuron rates, and reduced effective rank. The addition of SEM mitigates this instability. Experiments are averaged over 3 independent seeds, with shaded areas reporting 95% confidence intervals.

**A demonstration on CIFAR-10.** We illustrate this phenomenon with a toy experiment on CIFAR-10 (Krizhevsky, 2009). We compare two training regimes: (i) a stationary setting with fixed labels, and (ii) a non-stationary setting where labels are periodically shuffled to mimic *the bootstrap dynamics* of deep RL (Castanyer et al., 2025). Let $(x, y)$ be training samples with $y \in \{1, \ldots, K\}$. In the stationary regime, targets are fixed, so the conditional distribution $p(y|x)$ is constant and the empirical risk minimizer $\theta_t^\star$ remains stable up to stochastic fluctuations. In the non-stationary regime, labels are periodically shuffled so that $y \mapsto \pi_t(y)$, where $\pi_t$ is a permutation applied every $T$ steps. This induces inflection points in the minimizer, shifting whenever $\pi_t$ changes. Fig. 1 shows that in the stationary regime, training is stable: losses decrease smoothly, dormant neuron rates remain low, and effective rank increases, indicating robust representation learning (Dohare et al., 2024; Sokar et al., 2023; Liu et al., 2025c). In contrast, in the non-stationary regime, we observe instability: oscillating losses, rising neuron dormancy, and collapsing feature rank. Even in this simple supervised setting, instability in the target distribution alone is sufficient to undermine representational integrity. Similar optimization instabilities have been observed in prior work when evaluating CIFAR-10 under non-stationary conditions (Igl et al., 2021; Lee et al., 2023; Galashov et al., 2024).

**Stabilizing Representations under Non-Stationarity with SEM** *Simplicial Embeddings (SEM)* can mitigate this effect by projecting features onto a structured space that prevents collapse. The transformation enforces energy preservation; since each block has unit mass, representations cannot vanish and $\mathrm{tr}(\Sigma_t)$ remains bounded away from zero. It also promotes diversity, as intra-block competition spreads information across coordinates, while multiple blocks $(L)$ increase effective rank, counteracting covariance deflation. As shown in Fig. 1, critics trained with SEM retain higher effective rank, larger gradient energy, and lower neuron dormancy even when targets drift.

> **Takeaways:**
> - Non-stationarity exacerbates representation collapse, as evidenced by increased neuron dormancy and reduced effective rank.
> - Simplicial Embeddings (SEM) introduce a simplex-based geometric prior that sustains feature diversity and prevent feature collapse.

## 4 UNDERSTANDING THE IMPACT OF SEM ON DEEP RL NETWORKS

In actor–critic methods such as FastTD3, the critic is trained against bootstrapped targets $y_t(s, a) = r(s, a) + \gamma Q_{\phi^-}(s', \pi_\theta(s'))$. Both the target distribution $\mathcal{D}_t$ (samples $(s, a, r, s')$ from the replay buffer) and the target value $y_t$ evolve as the policy $\pi_\theta$ is updated. This continual drift produces a persistent bias term in $b_t = \nabla \mathcal{L}_{t+1}(\theta_t^\star) = \mathbb{E}_{(s,a) \sim \mathcal{D}_{t+1}} \Big[ \big( Q_\phi(s, a) - y_{t+1}(s, a) \big) \nabla_\theta Q_\phi(s, a) \Big]$, which is nonzero whenever $\pi_\theta$ or $\mathcal{D}_t$ changes. Thus, the critic is never optimizing a fixed objective but is instead forced to chase a moving target.

Representation collapse under such non-stationarity poses a fundamental barrier to stable and efficient deep RL (see App. A for additional context). Standard actor–critic methods are particularly

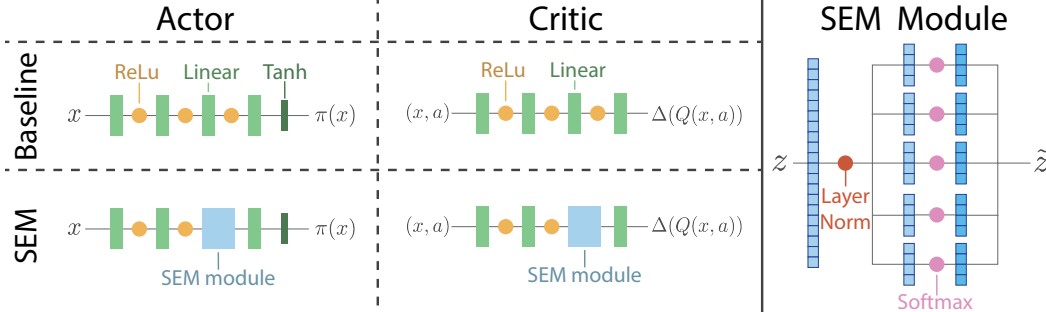

Fig. 2: **Actor–critic network architecture with SEM.** The actor (left) and critic (middle) architectures are modified with a SEM module, which partitions features into groups and applies group-wise softmax (right panel), constraining them to a product of simplices.

vulnerable. The critic's representations are trained against drifting targets, and the actor in turn depends on those representations to update its policy. This tight coupling amplifies instability, leading to poor sample efficiency in continuous control tasks. To address this challenge, we evaluate *Simplicial Embeddings (SEM)* as a representation-level regularizer. SEM aims to encourage the hidden features of both actor and critic networks to maintain a well-structured geometric organization, preventing collapse and preserving diversity.

**Setup.** Because this section involves a large number of ablations and is computationally expensive, we restrict experiments to five benchmarks from the Humanoid suite (Sferrazza et al., 2024), evaluated on (Seo et al., 2025). These tasks share the same robot-state dimension. We report aggregate performance across the five tasks and six seeds, with shaded areas reporting 95% confidence intervals, and provide full details in App. F.

**Integrating SEM on Actor-Critic Algorithms.** We choose FastTD3 (Seo et al., 2025), as our primary testbed. FastTD3 is specifically designed to be a simple and compute-efficient baseline for continuous-control and humanoid benchmarks. Its streamlined architecture yields strong performance while significantly reducing wall-clock training time. At the same time, FastTD3 inherits the critic-driven weaknesses of TD3; its bootstrapped value targets are generated online by the actor, making the critic susceptible to non-stationarity. This coupling amplifies representation collapse, as instabilities in the critic propagate to both value estimates and policy updates. We conduct most of our ablations on FastTD3, while later sections demonstrate that the benefits of SEM also extend to other actor–critic algorithms, such as SAC (Haarnoja et al., 2018) and PPO (Schulman et al., 2017).

SEM can either be applied to the actor, the critic, or both. We build on prior work showing that the penultimate layer plays a critical role in representation quality (Ceron et al., 2024c; Sokar & Castro, 2025), and that regularizing this layer can yield substantial performance gains. Fig. 2 illustrates how SEM is integrated into the actor–critic networks of FastTD3. For the critic, SEM replaces the baseline linear head with a structured projection, regularizing value estimates in the distributional C51 setting. For the actor, SEM is applied at the penultimate layer before the final linear+tanh, ensuring that the policy is conditioned on bounded and sparse features. Across the paper, dashed blue (**blue**, - -) curves indicate the baseline, while solid green, (**green**, —) curves represent the interventions added to the baseline. Fig. 3 shows clear gains when applying SEM to the actor or to both actor and critic, and more moderate gains when applied only to the critic. Although different SEM dimensions ($V$) improve sample efficiency and asymptotic performance, $V = 64$ appears most effective. We further explore the relationship between $L$ and $V$ (see sec 4), as this tradeoff was a central focus of the original SEM study (Lavoie et al., 2023). These results echo the non-stationary CIFAR-10 experiment, where SEM prevented feature collapse and stabilized learning (see Fig. 1).

**The Effect of SEM on Learning Dynamics in Deep RL.** We empirically evaluate the impact of SEM on the stability and efficiency of actor–critic algorithms. Our analysis combines both *learning performance* (returns, losses, TD error, critic disagreement) and *representation quality* (effective rank, feature norms), allowing us to connect sample-efficiency gains to underlying representational dynamics. This dual perspective highlights not only *whether* SEM improves performance, but also *why* it stabilizes training. A detailed explanation of each metric is provided in App. G.

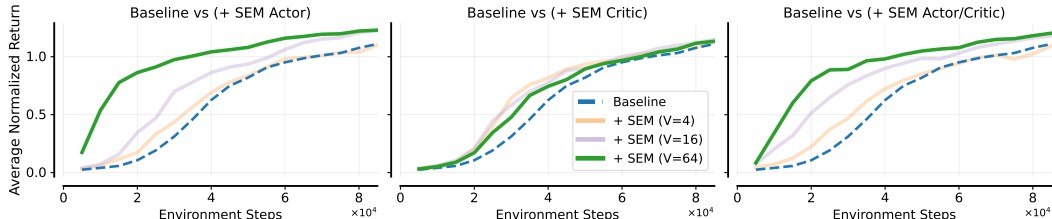

Fig. 3: **Average normalized return on** $5$ **HumanoidBench tasks over** $6$ **seeds.** Baseline agent (**blue**, - -) vs. SEM variants applied to actor, critic, or both. Each curve corresponds to an embedding dimension; $dim = 64$ (**green**, —) is highlighted. SEM accelerates early learning and improves asymptotic performance, with $dim = 64$ giving the most stable gains. In the three figures we use $L = 2$ for the SEM module.

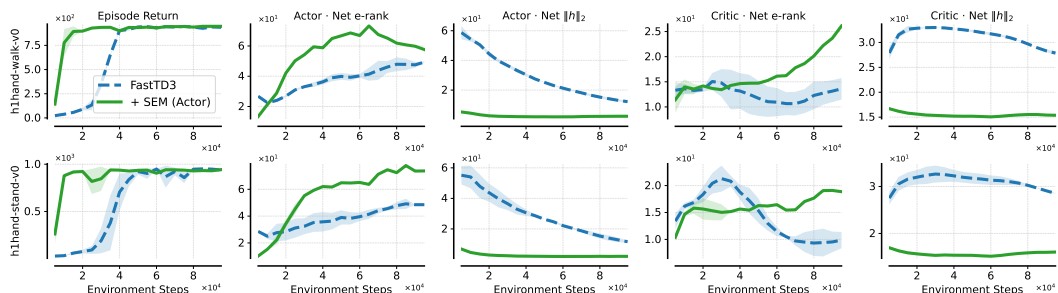

Fig. 4: **Learning and representation diagnostics on** $2$ **HumanoidBench tasks over** $6$ **seeds.** SEM reaches high return earlier, raises actor/critic effective rank, and keeps actor features compact.

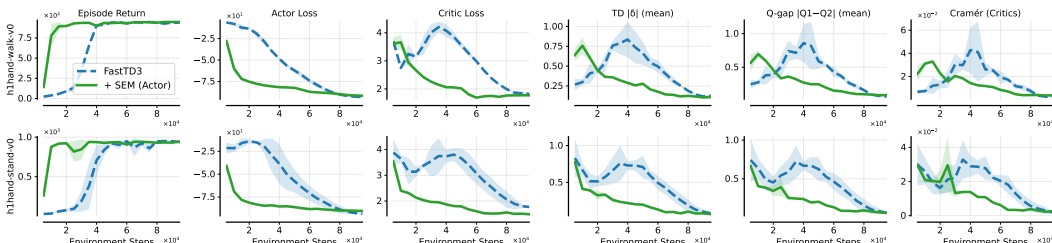

Fig. 5: **Learning dynamics on** $2$ **HumanoidBench tasks.** SEM reaches high return faster, with lower losses, smaller TD error, reduced critic disagreement, and better-calibrated value estimates.

To understand *why* SEM improves performance, we turn to representation-level diagnostics. Fig. 4 shows that SEM increases the effective rank of actor features, and bounds actor feature norms. Late in training, SEM also lifts the critic effective rank, a signs of more expressive and robust value learning. High effective rank is a proxy for avoiding representational collapse (Moalla et al., 2024; Mayor et al., 2025). In the deep RL literature, representation collapse under drift has been empirically associated with capacity loss (Lyle et al., 2021), deterioration of feature rank (Kumar et al., 2021b), and implicit under-parameterization (Kumar et al., 2021a). In supervised and self-supervised settings, techniques like orthogonality regularization and rank-preserving weight regularizers are used to prevent feature collapse (He et al., 2024). These representational patterns align with our formal analysis, showing that SEM prevents covariance deflation and sustains gradient energy, thereby preventing feature collapse and boosting performance.

As shown in Fig. 5, SEM improves optimization stability over the baseline. Agents with SEM achieve higher returns earlier and maintain smaller, more stable TD errors, reduced critic disagreement, and lower critic-distribution discrepancy. Such effects are crucial, as instability in bootstrapped critics is a primary failure mode of actor–critic methods (Fujimoto et al., 2019; Kumar et al., 2021a). By constraining representation geometry, SEM produces better-conditioned features that yield more calibrated value estimates, echoing similar findings in representation regularization for deep RL (Anand et al., 2019; Laskin et al., 2020; Schwarzer et al., 2021). These results indicate that SEM not only accelerates learning but also yields more calibrated value estimates, mitigating instability in bootstrapped critics.

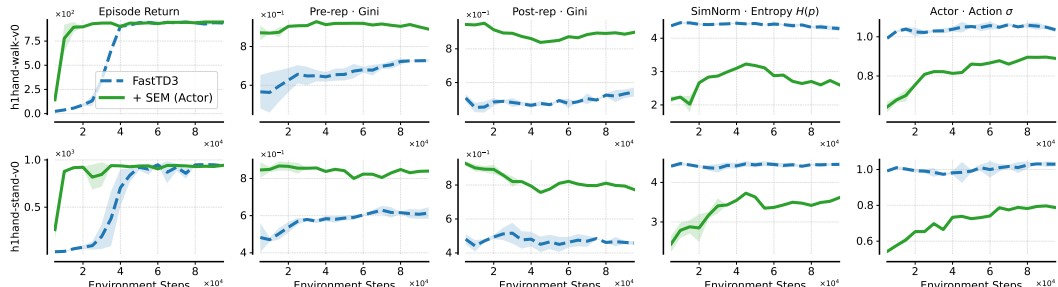

Fig. 6: **Sparsity, entropy, and action std on** $2$ **HumanoidBench tasks.** SEM agents achieve higher returns with sparser features, lower entropy, and more stable action scales.

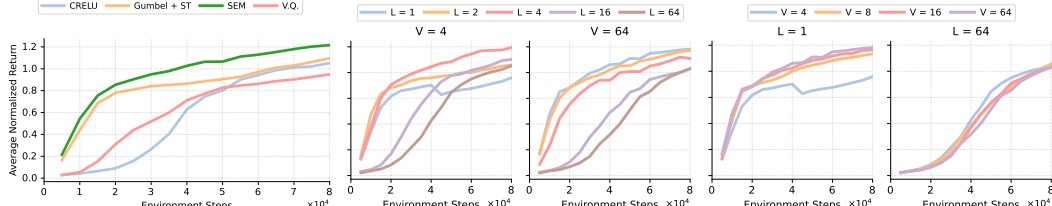

Fig. 7: **Aggregated average return on 5 HumanoidBench tasks.** We constrain the encoder's output of the actor. (left) SEM outperforms alternative methods to impart structure on the encoder's output. In SEM, representation capacity scales with $L \times V$ since the embedding consists of $L$ simplex groups of size $V$. When varying $L$ with fixed $V$ (middle panels), performance improves as $L$ increases from low-capacity regimes, and then saturates once $L \times V$ is sufficiently large. When varying $V$ with fixed $L$ (right panels), we observe the same pattern: for small $L$ (e.g., $L=1$), increasing $V$ noticeably improves performance, whereas for large $L$ (e.g., $L=64$), all values of $V$ perform similarly since the model already operates in a high-capacity regime.

In Fig. 6, we focus our lens on the SEM module itself and examine how it shapes representations and action behavior. As training proceeds, the SEM layer's activations become markedly sparser (higher Gini (Hurley & Rickard, 2009; Zonoobi et al., 2011)) and more sharply peaked (lower simplex entropy), while the overall action variance from the policy also declines. This trend is consistent with SEM's design, where the block-wise softmax promotes competition and selective activation. As a result, the module imposes structured, energy-preserving constraints on its layer, encouraging more decisive feature usage and reducing noise in the downstream policy mapping. Interestingly, this pattern also resonates with prior work in deep RL and representation learning. Hernandez-Garcia & Sutton (2019) show that enforcing sparsity in representations can improve robustness and mitigate interference in Q-learning settings. Moreover, recent studies on sparse architectures in deep RL such find that appropriately structured sparsity can enhance training stability and efficiency (Graesser et al., 2022; Ceron et al., 2024c;b; Ma et al., 2025).

**Comparing SEM to other Regularization Methods** To contextualize the benefits of simplicial embeddings, we compare SEM to alternative methods to induce structure on the encoder's output. We compare SEM to commonly used methods for learning discrete explicit representations: Gumbel + straight-through (Jang et al., 2017; Maddison et al., 2017) and Vector Quantization (van den Oord et al., 2017). We also compare SEM to C-RELU (Abbas et al., 2023) which have been shown to improve the representation's stability. We present the results in Fig. 7 (left) and find SEM to be more efficient and to lead to higher return than alternative methods. We conjecture that such improvement over Gumbel + ST and Vector quantization can be attributed to the fact that SEM does not necessitate the use of the straight-through estimator.

**Analyzing SEM Parameters in Deep RL** Lavoie et al. (2023) highlighted the effect of the simplex dimensionality $V$ and number of simplices $L$, which jointly control sparsity and capacity of the representation. Investigating these parameters in deep RL is essential to understand how SEM balances representation capacity and stability under non-stationary training, and whether the same tradeoffs observed in self-supervised representation learning extend to RL. We study the effect of varying $V$ and $L$ in Fig. 7 (middle and right, respectively). Our results show that performance is

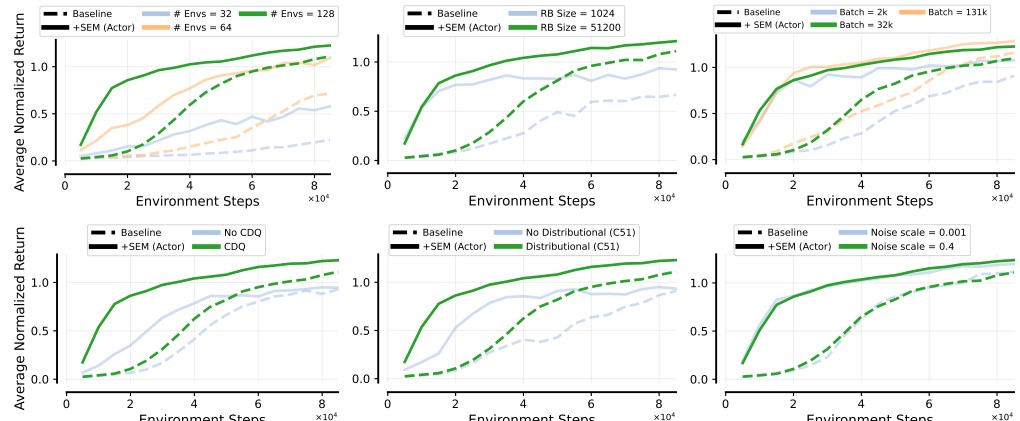

Fig. 8: **Effect of core design choices on FastTD3 with and without SEM on** 5 **HumanoidBench tasks.** SEM solid green, (**green, —**) consistently improves sample efficiency and asymptotic return across all settings, showing robustness to both hyperparameter and architectural design choices.

driven primarily by the total representational capacity $L \times V$. In low-capacity regimes (e.g., $L = 1$), increasing $V$ provides clear gains, consistent with the need for additional expressive power. However, once capacity is sufficiently large (e.g., $L = 64$), the effect of $V$ largely saturates, different choices of $V$ yield nearly identical performance; and in some cases smaller $V$ (e.g., $V = 4$) performs slightly better. Similarly, increasing $L$ improves performance when $L \times V$ is small, but the gains taper off once the model enters a high-capacity regime.

**FastTD3 Design Choices and Simplicial Embeddings.** FastTD3 extends TD3 with several design choices that improve throughput and stability, including parallel simulation, large-batch training, and distributional critics (Seo et al., 2025). These modifications enable actor–critic learning to scale efficiently in wall-clock time, but they do not address the geometry of the learned representations. In this section, we analyze how SEM complements FastTD3 by regularizing representation space and evaluate its effectiveness across the algorithmic design choices. In Fig. 8, we observe that SEM outperforms the baseline even when the agent is trained with reduced data availability (fewer environments, smaller replay buffers, or smaller batch sizes). Comparable gains also appear when algorithmic design choices such as CDQ and C51 are removed. These results demonstrate the robustness of SEM across both data-limited and simplified agent settings.

## 5 EMPIRICAL EVALUATION

We further evaluate the effectiveness and generality of SEM across a diverse set of deep RL algorithms and environments. Our study spans both off-policy and on-policy methods, including FastTD3, FastTD3-SimBaV2, FastSAC (Seo et al., 2025), and PPO (Schulman et al., 2017). Experiments are conducted on challenging humanoid benchmarks (28-h1hand tasks), (Sferrazza et al., 2024), Isaac Lab (Mittal et al., 2023), Isaac Gym suite (Makoviychuk et al., 2021), MTBench (Joshi et al., 2025), and an extended Atari-10 setup comprising 28 ALE games (see D.0.3 for more details) (Bellemare et al., 2013; Aitchison et al., 2023; Fedus et al., 2020), covering both continuous-control and pixel-based settings. Following prior work (Seo et al., 2025; Castanyer et al., 2025), we evaluate continuous-control tasks with six seeds and Atari results with three seeds, and aggregate performance across environments is reported, with shaded areas representing 95% confidence intervals. Full environment details and hyperparameter configurations are provided in App. I.

**Fast Actor–Critic Algorithms.** We first evaluate SEM on the HumanoidBench benchmark using three recent fast actor–critic baselines: FastTD3, FastTD3–SimBaV2, and FastSAC (Seo et al., 2025). These algorithms represent compute–efficient variants of TD3 and SAC, designed to scale with parallel simulation while maintaining strong performance on high–dimensional humanoid control. FastTD3–SimBaV2 incorporates hyperspherical normalization and reward scaling to accelerate critic training and stabilize optimization (Lee et al., 2025b); and FastSAC adapts the entropy–regularized SAC framework with similar throughput–oriented design choices, achieving high paral-

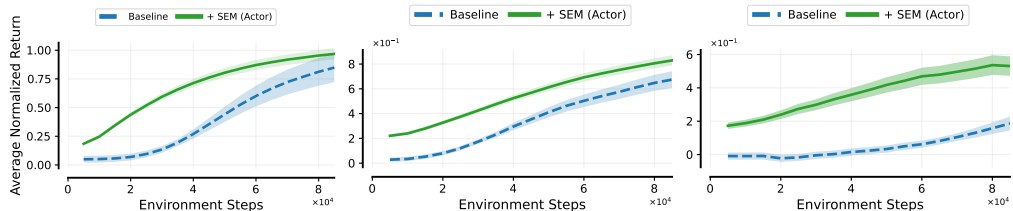

Fig. 9: **SEM on fast actor–critic algorithms.** Average normalized return on HumanoidBench with FastTD3 (left), FastTD3–SimBa (middle), and FastSAC (right). SEM, solid green, (**green**, —), consistently improves sample efficiency and yields higher final performance across all algorithms.

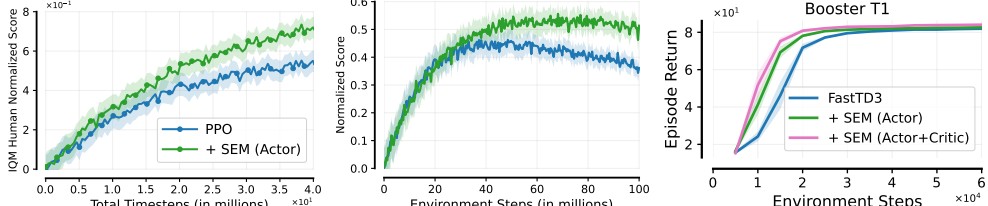

Fig. 10: **Performance of PPO with and without SEM across tasks.** Atari experiments on an extended Atari-10 setup (28 ALE games; pixel-based) (left). PPO in Isaac Gym (middle). Booster T1 (humanoid robot with fixed arms) comparing FastTD3. Applied SEM accelerates learning (right).

lel efficiency while preserving training stability. Across all three baselines, integrating SEM into the actor consistently accelerates early learning and improves asymptotic return. As shown in Fig. 9, SEM agents not only converge faster than their respective baselines, but also maintain lower variance across seeds. These results demonstrate that SEM provides complementary benefits to fast actor–critic methods, enhancing both stability and sample efficiency without modifying their underlying optimization procedures (see App. J for per-task learning curves). Fig. 10 (*right*) further shows that SEM improves policy learning on the `Booster T1` humanoid robot (Seo et al., 2025), a real-robot benchmark used to validate transfer from large-scale MuJoCo training (Zakka et al., 2025) (see D.0.5 for more details). We also evaluate FastTD3 on `12-h1`, `12-g1` tasks and `9-Isaac Gym` tasks, where a similar pattern is observed, as shown in App. K.

**Proximal Policy Optimization Algorithm.** To evaluate the generality of SEM beyond off-policy methods, we integrate it into PPO (Schulman et al., 2017), a popular on-policy method, using the CleanRL implementation (Huang et al., 2022). We evaluate SEM on two distinct benchmarks, Isaac Gym for continuous control and the ALE (Bellemare et al., 2013) for pixel-based discrete control in Atari games. In both domains, SEM improves PPO by accelerating convergence and increasing final returns. The per-environment learning curves are shown in Fig. 30. Aggregate results are summarized in Fig. 10, with the left panel showing performance gains on the ALE suite[2], and the middle panel showing improvements on the Isaac Gym tasks. These results demonstrate that SEM's benefits are not limited to TD3-style critics but extend to policy-gradient methods and vision-based RL, underscoring its broad applicability.

**Multitask and Offline-to-online Deep RL.** Recent work by Joshi et al. (2025) introduced a large-scale benchmark for multi-task reinforcement learning (MTRL) in robotics. Implemented in Isaac Gym, this benchmark comprises over seventy robotic control problems spanning both manipulation and locomotion, with subsets such as MT50 focused on manipulation. We compare FastTD3 (Seo et al., 2025) to its SEM-augmented variants (+SEM). As shown in Fig. 10 (*right*), +SEM improves sample efficiency, achieving faster learning and higher returns within the same training budget. We also evaluate Flow Q-Learning (FQL; Park et al., 2025b), an actor–critic–style method that couples a value function with a flow-based policy generator trained to transport actions toward high-value regions. Despite differing from standard policy-gradient updates, FQL preserves the critic-guided policy improvement structure central to actor–critic algorithms. Applying SEM to the flow-based actor improves sample efficiency and stability during the offline-to-online transition (see Fig. 11),

---

[2]For the SEM module, we use $L = 128$ and $V = 4$ when evaluating ALE games (Bellemare et al., 2013) with PPO, whose penultimate fully connected layer has dimensionality 512.

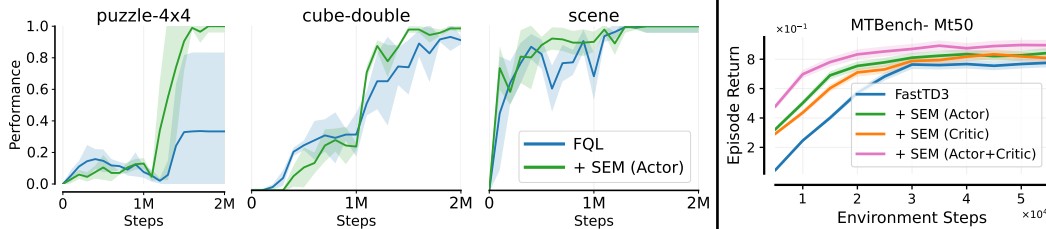

Fig. 11: **Left:** Offline-to-online RL results on 3 OGBench tasks (5 seeds) (Park et al., 2025a). Online fine-tuning starts at 1M steps. **Right:** MTBench MT50 (robotics tasks) comparing FastTD3.

where representations must adapt from static replay data to evolving on-policy distributions. We view these observations as opening opportunities to investigate whether simplex-constrained embeddings can help mitigate representation drift and facilitate adaptation; establishing this more broadly will require further study and is left to future work.

**Value-Based Deep RL.** Although our evaluation centers on actor–critic methods, this choice reflects experimental scope rather than a conceptual limitation. SEM operates on learned representations and is compatible with value-based algorithms. Extending SEM to these settings, such as DQN-style architectures (Mnih et al., 2013), is an important avenue for improving sample efficiency. To explore this direction, we evaluated SEM within PQN (Gallici et al., 2025), a recently proposed value-based algorithm that simplifies deep temporal-difference learning by streamlining target computation and reducing unnecessary complexity. Our preliminary PQN experiments indicate that, while SEM yields improvements in a few games (see Fig. 37), it does not provide consistent gains in this regime overall (see Fig. 38). This suggests that a more systematic investigation is needed to determine when geometric constraints benefit value-based methods. See App. M for more details.

## 6 DISCUSSION

Our results demonstrate that geometric priors on representation space can substantially improve the efficiency of deep RL agents. By constraining features to a product of simplices, SEM yields bounded and sparse embeddings that avoid feature collapse and neuron dormancy under non-stationarity. This lightweight inductive bias requires no auxiliary losses, adds effectively zero computational cost (see Table 2), and consistently improves sample efficiency and asymptotic return across actor–critic methods and diverse benchmarks. Unlike existing model-based RL approaches using discrete state embeddings (Hansen et al., 2024; Hafner et al., 2020; 2025; Scannell et al., 2025), SEM does not require auxiliary objectives or additional networks. Surprisingly, its benefits are most pronounced when applied to the actor's penultimate layer, where feature geometry most directly shapes policy gradients. Our analyses indicate that SEM alleviates several optimization difficulties in deep RL (Moalla et al., 2024; Juliani & Ash, 2024). By preserving effective rank, bounding feature norms, and reducing critic disagreement, SEM provides more reliable gradients and stabilizes the bootstrapped targets that often undermine critic training. These effects highlight representation geometry as a simple but powerful lever for stabilizing learning under non-stationarity.

**Limitations and Future Work.** SEM is not a universal remedy. In tasks with extreme distribution shift or very sparse rewards, feature collapse and critic drift may still occur, and SEM introduces hyperparameters $(L, V, \tau)$ that require light tuning to balance sparsity and capacity. For consistency and computational efficiency, we adopted the baseline models' default hyperparameters across all experiments. Nonetheless, RL agents are notably sensitive to these choices (Ceron et al., 2024a; Patterson et al., 2024), and ideally each experimental setting would undergo a dedicated hyperparameter search, though this is often computationally prohibitive. Moreover, our experiments focus on continuous control and Atari; its impact on large-scale vision or language-conditioned RL remains untested. Future work should investigate adaptive schedules for $(L, V, \tau)$, and integration in more general-purpose algorithms such as MR.Q (Fujimoto et al., 2025), which combine multiple objectives and scale across domains. Another direction is to examine whether SEM benefits value-based algorithms, and to explore both its potential for scaling network architectures (Ceron et al., 2024b; Sokar et al., 2025) and its interaction with architectural priors (e.g., MoEs, Residual Nets) (Ceron et al., 2024c; Castanyer et al., 2025; Kooi et al., 2025).

## ACKNOWLEDGMENTS

The authors would like to thank Ali Saheb Pasand and Lu Li for valuable discussions during the preparation of this work. We thank Younggyo Seo for his kindness in answering questions about the FastTD3 repository. Gopeshh Subbaraj deserves a special mention for providing valuable feedback on an early draft of the paper.

The research was enabled in part by computational resources provided by the Digital Research Alliance of Canada (https://alliancecan.ca) and Mila (https://mila.quebec). Pablo Samuel Castro acknowledges funding from NSERC Discovery Grant. We want to acknowledge funding support from Google and CIFAR AI. We would also like to thank the Python community (Van Rossum & Drake Jr, 1995; Oliphant, 2007) for developing tools that enabled this work, including NumPy (Harris et al., 2020), Matplotlib (Hunter, 2007), Jupyter (Kluyver et al., 2016), and Pandas (McKinney, 2013).

## ETHICS STATEMENT

This paper presents work whose goal is to advance the field of Machine Learning, and reinforcement learning in particular. There are many potential societal consequences of our work, none which we feel must be specifically highlighted here.

## REPRODUCIBILITY STATEMENT

We provide all the details to reproduce our results in the Appendix.

## LLM USE

LLMs were used to assist paper editing and to write the code for plotting experiments.

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

# APPENDIX CONTENTS

# A    RELATED WORK

**Stability in Deep RL**    A longstanding challenge in reinforcement learning is the stability of value-based updates in actor–critic methods. One major source of instability is overestimation bias, which accumulates when bootstrapped critics reinforce overly optimistic targets. Twin Delayed DDPG (TD3) (Fujimoto et al., 2018) mitigates this issue with clipped double Q-learning and delayed policy updates, producing more reliable critics and improving control performance. Another direction seeks to stabilize targets by modeling full return distributions rather than point estimates. Distributional RL methods such as C51 (Bellemare et al., 2017), QR-DQN (Dabney et al., 2018b), and IQN (Dabney et al., 2018a) show that capturing the shape of the return distribution reduces variance and provides richer learning signals.

More recently, architectural choices have been used to enhance critic stability. SimBaV2 (Lee et al., 2025b) biases networks toward simpler, well-conditioned representations via input normalization, linear residual paths, and feature normalization, helping large models avoid divergence. Meanwhile, BRO shows that scaling critic capacity paired with strong regularization and optimistic updates yields dramatically better sample efficiency in continuous control tasks (Nauman et al., 2024b). In parallel, Ceron et al. (2024b;c); Ma et al. (2025) demonstrate that applying static network sparsity unlocks scaling of deep RL models by mitigating gradient interference (Bengio et al., 2020; Obando Ceron et al., 2023) and plasticity loss (Lyle et al., 2023). Training regimens also play a role; SR-SPR (D'Oro et al., 2022) demonstrates that periodic network resets counteract bootstrapping drift, allowing agents to sustain extremely high replay ratios without collapse. FastTD3 (Seo et al., 2025) integrates several of these lessons, combining parallel simulation, large-batch updates, and distributional critics to achieve strong stability at high throughput. Our approach is complementary to these efforts. Rather than modifying update schedules or ensemble targets, we constrain the geometry of latent representations, aiming to reduce critic variance and stabilize bootstrapped updates through structured embeddings.

**Sample-Efficient RL**    Beyond stability, a parallel line of work targets sample efficiency, with progress spanning both representation-driven methods and algorithmic or model-based improvements. Representation learning has emerged as a powerful way to extract more information per interaction. CURL (Laskin et al., 2020) applies contrastive learning to enforce invariances in encoders trained jointly with the control objective, significantly narrowing the gap between pixel- and state-based agents. SPR (Schwarzer et al., 2021) extends this idea with self-predictive latent dynamics, ensuring temporal consistency and yielding state-of-the-art data efficiency on Atari. Building on SPR, SR-SPR (D'Oro et al., 2022) adds scheduled resets that prevent drift and enable aggressive replay-ratio scaling. Other works inject architectural or learning biases for example; SimBa (Lee et al., 2025a) introduces simplicity constraints that regularize feature representations, allowing larger networks to remain well-conditioned under nonstationary training; its successor, SimBaV2 (Lee et al., 2025b), further stabilizes scaling through hyperspherical normalization, constraining weights and activations to unit-norm manifolds and ensuring consistent gradient magnitudes across capacities. Neuroplastic Expansion (Liu et al., 2025a) complements these structural interventions by dynamically growing and pruning neurons to preserve long-term adaptability, while The Courage to Stop (Liu et al., 2025b) improves behavioral efficiency by terminating unproductive trajectories early, reducing replay-buffer contamination.

For SALE (Fujimoto et al., 2023) further enriches the representation space with state–action embeddings, producing TD7, which substantially outperforms TD3 in continuous control. Outside of RL, simplicial embeddings (Lavoie et al., 2023) show that constraining features to products of probability simplices induces sparse, group-structured representations that generalize effectively in supervised and self-supervised settings and leads to a compositional representation (Ren et al., 2023; Lavoie et al., 2025). We draw inspiration from this idea and adapt it to reinforcement learning, inserting simplicial modules into fast actor–critic pipelines.

Algorithmic and model-based approaches provide another path to efficiency. Soft Actor-Critic (SAC) (Haarnoja et al., 2018) introduces maximum-entropy RL, balancing reward and exploration to achieve robust and data-efficient learning in continuous control. Model-based algorithms further improve efficiency by planning with learned dynamics. TD-MPC2 (Hansen et al., 2024) demonstrates that latent-space model predictive control scales effectively across diverse domains, achieving state-of-the-art performance with a single set of hyperparameters. EfficientZero (Ye et al., 2021) combines

MuZero-style search with learned latent dynamics, reaching human-level Atari performance with orders of magnitude fewer environment steps. Our method differs from these approaches by focusing on representation geometry: rather than auxiliary losses, ensembles, or world models, we show that a single simplicial bottleneck can consistently improve the sample efficiency of fast actor–critic algorithms while preserving their hallmark wall-clock advantages.

**Structured representation in RL**   Constraining the encoder's output is common in RL. C-ReLU has been shown to improve training and plasticity (Abbas et al., 2023). Feature normalization with L2 regularization of the features also improves training scalability and enables larger scale training of RL models. Closer to our work, DreamerV2 (Hafner et al., 2020) and DreamerV3 (Hafner et al., 2025) encode the observation into a one-hot discrete representation work. Scannell et al. (2025) also learn discrete latent space via a learned codebook and gumbel softmax with straight-through estimator. Wabartha & Pineau (2024) also propose to learn discrete encoding of the state for policy learning and show interpretable representations. However, methods with explicit discretization necessitate the use of a biased gradient estimator to propagate the learning signal inside the encoder. Similar to our work, Hansen et al. (2024) constrain the encoder's output into SEM. In this work, we find that SEM is a crucial component for improving sample efficiency and performance in RL and study that component in details and connect the improved performance to the improved training stability coming from the sparse and structured representation endowed by SEM.

## B   FORMAL ANALYSIS

**Theorem 1.** *Non-stationarity increases neuron dormancy.*

*Proof.* Let $\mathcal{D}_t$ be the data distribution at iteration $t$ and consider a critic $f_\theta(x) = W h_\phi(x)$, trained by minimizing the (mean) squared error to targets $y_t(x)$:

$$\mathcal{L}_t(\theta) = \mathbb{E}_{x \sim \mathcal{D}_t}\left[\left(f_\theta(x) - y_t(x)\right)^2\right]. \tag{3}$$

Define the minimizer $\theta_t^\star \in \arg\min_\theta \mathcal{L}_t(\theta)$ and tracking error $e_t = \theta_t - \theta_t^\star$. A first–order expansion of SGD around $\theta_t^\star$ gives

$$e_{t+1} \approx (I - \alpha H_t)\, e_t - \alpha\, b_t, \quad H_t = \nabla^2 \mathcal{L}_t(\theta_t^\star),\ b_t = \nabla \mathcal{L}_{t+1}(\theta_t^\star), \tag{4}$$

where $b_t = 0$ if $\mathcal{D}_{t+1} = \mathcal{D}_t$, but $b_t \neq 0$ under drift. This shows that the optimizer must continually track a moving minimizer, which destabilizes learned features. Let $z = h_\phi(x) \in \mathbb{R}^d$ with covariance

$$\Sigma_t = \mathrm{Cov}_{x \sim \mathcal{D}_t}(z) = \mathbb{E}[zz^\top] - \mathbb{E}[z]\mathbb{E}[z]^\top, \qquad \mathrm{srank}(\Sigma_t) = \frac{\|\Sigma_t\|_F^2}{\|\Sigma_t\|_2^2}. \tag{5}$$

In the stationary case, $\Sigma_t \to \Sigma$ with a large stable rank, preserving feature diversity. Under non-stationarity, the drift term in equation 4 induces oscillations in $\Sigma_t$ and systematic *covariance deflation* (drop in srank), a hallmark of collapse. When representations collapse (covariance deflation; equation 5), feature energy shrinks. For a linear head,

$$\mathbb{E}\left[\|\nabla_W \mathcal{L}_t\|_F^2\right] \leq 4\,\mathbb{E}[\delta_t^2]\ \mathrm{tr}(\Sigma_t), \qquad \delta_t = f_\theta(x) - y_t(x), \tag{6}$$

so smaller $\mathrm{tr}(\Sigma_t)$ directly yields smaller gradients and slower learning. With ReLU features $z = \sigma(a)$, the backprop signal through unit $j$ is gated:

$$\frac{\partial \mathcal{L}_t}{\partial a_j} = \mathbf{1}\{a_j > 0\}\, \langle \nabla_z \mathcal{L}_t, e_j \rangle \quad \Rightarrow \quad \mathbb{E}\left[\left\|\frac{\partial \mathcal{L}_t}{\partial a_j}\right\|^2\right] \leq p_{j,t}\, \mathbb{E}\left[\|\nabla_z \mathcal{L}_t\|_2^2\right], \tag{7}$$

where $p_{j,t} = \Pr(a_j > 0)$ and $e_j$ is the $j$-th basis vector. Non-stationary drift (equation 4) reduces $p_{j,t}$ and $\mathrm{Var}(z_j)$; together with lower $\mathrm{tr}(\Sigma_t)$, this shrinks per-unit updates and increases neuron dormancy (Sokar et al., 2023). $\qquad\square$

## C  FAST ACTOR–CRITIC

FastTD3 (Seo et al., 2025) extends the standard TD3 framework by combining *(i)* parallel simulation across many environment instances, *(ii)* large-batch critic updates, and *(iii)* algorithm design choices like distributional critics (C51) (Bellemare et al., 2017), noise scaling and clipped double Q-learning (CDQ) (Fujimoto et al., 2018).

Instead of approximating only the expected return, FastTD3 estimates the entire return distribution $Z(s,a)$ following C51 (Bellemare et al., 2017). The action–value distribution is approximated by a categorical distribution supported on $N$ fixed atoms $\{z_i\}_{i=0}^{N-1}$, uniformly spaced in $[v_{\min}, v_{\max}]$:

$$z_i \;=\; v_{\min} + i\,\Delta z, \qquad \Delta z \;=\; \frac{v_{\max} - v_{\min}}{N-1}, \quad i = 0, 1, \ldots, N-1.$$

The critic outputs logits $\{\ell_i(s,a)\}_{i=0}^{N-1}$ which define probabilities via $p_i(s,a) = \mathrm{softmax}(\ell(s,a))_i$. Given a transition $(s,a,r,s')$ and a next action $a' = \pi_{\mathrm{targ}}(s')$, the Bellman-updated support is

$$z_j' \;=\; \mathrm{clip}\big(r + \gamma z_j, \, v_{\min}, \, v_{\max}\big), \qquad j = 0, 1, \ldots, N-1,$$

with target probabilities $p_j(s',a')$ from the target critic at $(s',a')$. C51 projects this target distribution back onto the fixed support via the projection operator $\Phi$:

$$m_i \;\equiv\; (\Phi T Z)(z_i) \;=\; \sum_{j=0}^{N-1} p_j(s',a') \left[ 1 - \frac{|z_j' - z_i|}{\Delta z} \right]_+, \qquad [x]_+ \;=\; \max\{x, 0\}. \tag{8}$$

Training minimizes the cross-entropy between the projected target $m$ and the predicted distribution:

$$\mathcal{L}(s,a) \;=\; -\sum_{i=0}^{N-1} m_i \, \log p_i(s,a).$$

This distributional perspective reduces variance in target estimation and captures the multi-modality of returns in continuous control. Two other important stabilizers in actor–critic methods are *noise scaling* and *Clipped Double Q-learning (CDQ)*. Noise scaling injects Gaussian perturbations into the action to balance exploration and stability:

$$a = \pi_\theta(s) + \epsilon, \qquad \epsilon \sim \mathcal{N}(0, \sigma^2 I), \tag{9}$$

where the scale $\sigma$ must be tuned to avoid either poor exploration ($\sigma$ too small) or instability ($\sigma$ too large). On the other hand, CDQ mitigates overestimation bias by maintaining two critics $Q_{\phi_1}, Q_{\phi_2}$ and defining the bootstrapped target conservatively as

$$y = r + \gamma \min_{i=1,2} Q_{\phi_i^-}\big(s', \pi_\theta(s') + \epsilon\big), \qquad \epsilon \sim \mathcal{N}(0, \sigma_{\mathrm{target}}^2 I), \tag{10}$$

where $\phi_i^-$ are delayed target networks and $\sigma_{\mathrm{target}}$ controls target-smoothing noise. Each critic then minimizes its own squared Bellman error:

$$\mathcal{L}_Q(\phi_i) = \mathbb{E}_{(s,a,r,s') \sim \mathcal{D}} \left[ \big(Q_{\phi_i}(s,a) - y\big)^2 \right], \quad i = 1, 2. \tag{11}$$

Together, these design choices underpin the high-throughput yet stable behavior of FastTD3 (Seo et al., 2025).

## D  BENCHMARKS

### D.0.1  ISAAC GYM

For our experiments, we used the original Isaac Gym benchmark, which provides pre-built stand-alone environments and runs entirely on the GPU via a PhysX backend. This setup enables both physics simulation and neural network policy training on the GPU, offering high-throughput evaluation. Although Isaac Gym is deprecated, we used it to ensure reproducibility, specifically running the PPO algorithm from CleanRL on tasks spanning locomotion, robotic hands, and cube stacking (See Figure 12). To reproduce this task, we follow the PPO hyperparameters from CleanRL for Isaac, as presented in Table 3.

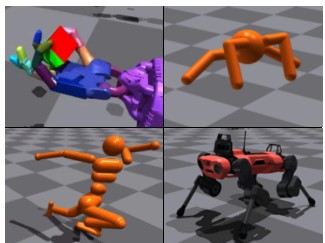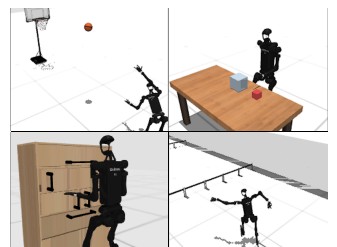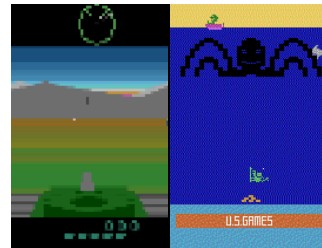

Fig. 12: **Environment Visualizations.** We evaluate SEM across three benchmark suites such as Isaac Gym, HumanoidBench, and Atari. The first two cover state-based locomotion/manipulation; Atari introduces pixel-based games of varying complexity.

### D.0.2 HUMANOIDBENCH

In our experiments, we used the Humanoid Benchmark, a suite of tasks for evaluating humanoid robot control across locomotion and manipulation, implemented on the MuJoCo physics engine. We focused on three robot configurations: the Unitree H1 without hands (26 DoF), the Unitree H1 with hands (76 DoF), and the Unitree G1 with three-finger hands (44 DoF). The benchmark defines 27 core tasks, and additionally, sit, balance, and bookshelf are each implemented in both simple and hard variants, while insert is implemented in small and normal configurations. This brings the total to 31 tasks. Our evaluations covered locomotion challenges, including walking, running, crawling, stair climbing, and balancing, and whole-body manipulation tasks such as opening doors, lifting packages, operating kitchen objects, and performing insertions. Together, these tasks provided a diverse and rigorous testing ground for our study of humanoid control (See Figure 12). To reproduce this task, we follow the fastTD3 hyperparameters (Seo et al., 2025), as presented in Table 4.

### D.0.3 ATARI

We conducted pixel-based reinforcement learning experiments using the Arcade Learning Environment (ALE) (Bellemare et al., 2013). We start from the Atari-10 suite (Aitchison et al., 2023) and extend it with additional environments from the Atari-20 suite (Fedus et al., 2020), yielding a broader subset of the ALE benchmark. In total, we evaluate 28 games spanning varying difficulty levels, trained with PPO from CleanRL (Huang et al., 2022) as the baseline. Two games overlap across the subsets (Bowling and Q*bert). We report IQM returns (see Figure 10, Left) following the evaluation protocol of (Agarwal et al., 2021; Ceron et al., 2024c;b; Sokar & Castro, 2025).

### D.0.4 MTBENCH

In our experiments, we used the Multi-Task Benchmark for Robotics, an open-source suite built on the GPU-accelerated Isaac Gym simulator. Specifically, we worked with the 50 manipulation tasks adapted from Meta-World, where a single-armed robot interacts with one or two objects through actions such as pushing, picking, and placing. Each task provides parametric variations in object initialization and target positions, adding diversity and complexity. For evaluation, we adopted the MT50 setting, which encompasses the full set of 50 tasks.

### D.0.5 BOOSTER GYM HUMANOID ROBOT

As reported in the FastTD3 paper (Seo et al., 2025), the authors configured the Booster Gym humanoid robot to operate within the MuJoCo Playground environment, which supports 12 degrees of freedom (DOF). They trained the policy entirely in simulation and successfully transferred it to a real robot, demonstrating an effective sim-to-real deployment of an off-policy RL method on a full-sized humanoid. In our experiments, we used the same robot model and hyperparameters as FastTD3, also within MuJoCo Playground. Our configuration led to faster convergence and improved performance in simulation compared to the reported baseline.

Table 1: Default hyper-parameters setting for actor-critic MLP

| Hyper-parameter | Value |
|---|---|
| Critic Hidden Dim | 1024 |
| Actor Hidden Dim | 512 |
| Critic Learning Rate | 3e-4 |
| Actor Learning Rate | 3e-4 |

# E  DEEP RL NETWORK ARCHITECTURES

### E.0.1  MLP

We modified the FastTD3 architecture, specifically in the actor–critic design, where both networks are implemented as multilayer perceptrons (MLPs). The critic receives concatenated observation–action inputs, while the actor processes only the observations. In both cases, the inputs first pass through two linear layers with ReLU activations. At this point, we introduced the SEM mechanism, which can be enabled or disabled, and applied selectively to the actor, the critic, or both. For the critic, if SEM is not used, the representation is processed by a sequence of `Linear→ReLU→Linear` layers, with the final linear layer outputting dimension *num_atoms*. If SEM is enabled, the sequence becomes `SEM→Linear`, again producing an output of size *num_atoms*. For the actor, the representation without SEM follows a `Linear→ReLU→Linear→Tanh` sequence, while with SEM it follows `SEM→Linear→Tanh`, where the final `Tanh` ensures bounded continuous actions. In Table 1, we present the fixed hyperparameters used across all environments. Other hyperparameters, such as *num_atoms* or *num_env*, varied depending on the environment, in which case we adopted the values proposed by (Seo et al., 2025).

### E.0.2  CNN

For our pixel-based experiments, we modified the PPO implementation from CleanRL, which follows an actor–critic design. The shared backbone consists of three convolutional layers, each followed by a ReLU activation, producing a flattened representation that is then processed by a two-layer MLP with ReLU activations. This representation is used by both the actor and the critic. In our intervention, we introduced the SEM block into the actor: when enabled, the representation passes through the SEM block before a final linear layer; when disabled, it follows a `Linear→ReLU→Linear` sequence. The critic remains unchanged, while the actor architecture is varied depending on the use of SEM. We adopted the PPO hyperparameters for Atari from CleanRL (Huang et al., 2022), as summarized in Table 5.

# F  ABLATION SETUP

Given our constrained computational budget, we performed experiments on a subset of HumanoidBench, consisting of five robotics tasks. These tasks are part of the benchmark evaluated with FastTD3 (Seo et al., 2025) and correspond to `h1hand-{walk, stand, run, stair, slide}-v0`. All ablation experiments were conducted on this subset using six random seeds (see Fig. 13 and Fig. 14).

# G  METRICS

To better understand the dynamics of training and the quality of learned representations, we report a diverse set of metrics beyond standard returns. These measures capture complementary aspects of learning, including representation diversity, network expressivity, parameter stability, gradient behavior and sampling efficiency. Results of these analyses are provided in section 4.

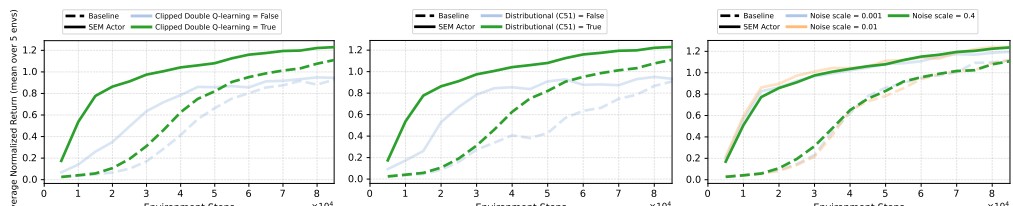

Fig. 13: **Effect of core hyperparameters.** SEM Actor compared to the baseline across (left) number of parallel environments, (middle) replay buffer size, and (right) batch size. SEM consistently scales better and achieves higher returns.

Fig. 14: **Robustness to design choices.** SEM Actor vs. baseline across (left) clipped double Q-learning, (middle) distributional critic (C51), and (right) exploration noise scale. SEM remains robust, while the baseline is more sensitive.

## G.1 FEATURE RANK

This metric assesses the quality of learned representations in deep RL by identifying the smallest subspace that retains 99% of the variance, thereby enhancing interpretability, efficiency, and stability. A higher feature rank indicates more diverse representations. The computation follows the approximate rank from (Yang et al., 2019; Moalla et al., 2024):

$$\sum_{i=1}^{k} \frac{\sigma_i^2}{\sum_{j=1}^{n} \sigma_j^2} \geq \tau,$$

where $\sigma_i$ are the singular values of the feature matrix, $n$ is the total number of singular values, and $\tau$ is the variance threshold (e.g., 99%). The feature rank $k$ is the smallest number of principal components required to preserve at least $\tau$ of the total variance.

## G.2 DORMANT NEURONS

This metric quantifies the proportion of neurons with near-zero activations, which limits the network's expressivity. It serves to detect inefficiencies in learning, as a high proportion of dormant neurons implies that many units are inactive or rarely contribute to the output. The computation follows (Sokar et al., 2023):

$$\frac{\sum_{i=1}^{N} \mathbf{1}(|a_i| < \epsilon)}{N} \times 100,$$

where $N$ is the total number of neurons, $a_i$ is the activation of neuron $i$, $\epsilon$ is a small threshold (e.g., $10^{-5}$), and $\mathbf{1}$ is the indicator function.

## G.3 WEIGHT NORM

This metric measures the magnitude of neural network weights, providing insight into model complexity, stability, and overfitting risk. Large weight norms indicate parameters with high magnitudes, which may hinder generalization. The metric is computed as in (Moalla et al., 2024; Lyle et al., 2021):

$$\|\theta\|_2 = \sqrt{\sum_i \theta_i^2},$$

where $\theta_i$ are the weights of a given layer.

### G.4 GINI SPARSITY

The Gini metric is used to quantify the sparsity of neural representations. A high Gini value indicates a sparse representation, where only a few neurons are strongly active while most remain near zero; this often improves interpretability, makes more efficient use of network capacity, and can help reduce overfitting. In contrast, a low Gini value corresponds to dense representations, where many neurons are active simultaneously, allowing the network to capture richer information but often at the cost of reduced interpretability and potentially noisier features. In practice, we observed a direct relationship between the Gini metric and the return when using SEM, with better performance associated with higher Gini values. The Gini value is computed using the following equation.

$$G \;=\; 1 + \frac{1}{n} \;-\; \frac{2}{n \sum_{i=1}^{n} v_i} \; \sum_{i=1}^{n} (n + 1 - i)\, v_{(i)}$$

where where

$$v = \big(|x_1|, |x_2|, \ldots, |x_n|\big)$$

It is the vector of all activations, taken in absolute value and stacked into one vector. The Gini metric has been explored in the papers (Hurley & Rickard, 2009; Zonoobi et al., 2011).

### G.5 CRAMER DISTANCE

The Cramér distance is defined as the squared $L_2$ distance between the cumulative distribution functions (CDFs) of two probability distributions. When the distributions are similar, their CDFs overlap closely and the Cramér distance approaches zero. Conversely, when the distributions differ, the CDFs diverge and the distance increases. In practice, a lower Cramér distance indicates that the learned distribution is closer to the target distribution, which is desirable. Empirical results also suggest a correlation between lower Cramér distance and improved returns. This measure is computed using the following equation:

$$D_{\text{Cramér}}^2(p_1, p_2) \;=\; \sum_{j=1}^{n} \Big( F_{p_1}(z_j) - F_{p_2}(z_j) \Big)^2 \Delta z$$

where $p_1$ and $p_2$ are probability distributions, and $F_{p_1}, F_{p_2}$ denote their corresponding cumulative distribution functions (CDFs).

### G.6 ENTROPY

This metric measures the average entropy of the representations. High entropy indicates that the representation is more dispersed, less concentrated, and carries more uncertainty. Low entropy corresponds to a more concrete representation, with higher sparsity. In practice, we observe a relationship where lower entropy is associated with better returns and a higher Gini measure. This metric is defined by the following equation;

$$p_{i,j} = \frac{p_{i,j}}{\sum_k p_{i,k} + \varepsilon}$$

$$\text{entropy} = \frac{1}{B} \sum_{i=1}^{B} \left( - \sum_j p_{i,j} \log(p_{i,j} + \varepsilon) \right)$$

where $B$ is the batch size, and $p$ is the non-negative representation normalized to form a probability distribution.

## H ADDITIONAL BASELINES

We expand our comparison to include a broader set of interventions commonly used to address plasticity loss, representation collapse, and optimization pathologies in deep RL. Following the setup

described in section 4 (*Comparing SEM to other Regularization Methods*), we evaluate eight additional methods: Reset (Nikishin et al., 2022), L2 regularization (Kumar et al., 2023), ReGen (Kumar et al., 2025), Dropout (Hendrycks, 2016), Shrink-and-Perturb (Ash & Adams, 2020), GELU activation (Hinton et al., 2012), Weight Clipping (Elsayed et al., 2024), and Spectral Normalization (Gogianu et al., 2021). Each method is evaluated across 6 random seeds and 5 HumanoidBench tasks.

These experiments are *in addition* to those reported in Fig. 7, which already include other relevant baselines such as Gumbel with straight-through estimation (Jang et al., 2017; Maddison et al., 2017), Vector Quantization (van den Oord et al., 2017), and C-ReLU (Abbas et al., 2023). We demonstrate that SEM offers a lightweight, stable, and effective mechanism for mitigating representation collapse in deep RL without requiring complex architectural modifications or extensive hyperparameter tuning.

## H.1   REGEN

We applied ReGen by zeroing tiny weights using three thresholds, $\tau = 10^{-5}, 10^{-6}$ and $\tau = 10^{-7}$, allowing us to compare how different magnitudes influence sparsity and model stability (Kumar et al., 2025).

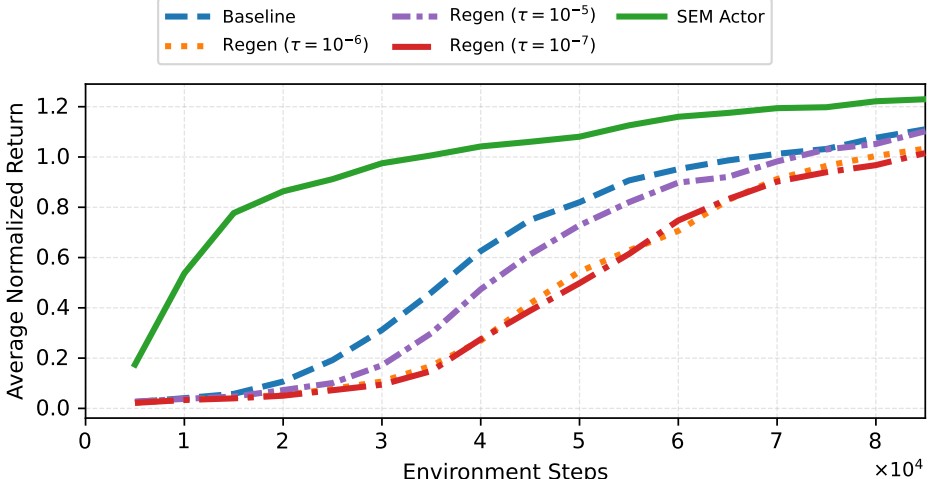

Fig. 15: **Average normalized return on 5 HumanoidBench tasks over 6 seeds.** We compare the Baseline and ReGen variants with SEM Actor. SEM variation consistently outperforms all ReGen thresholds, achieving faster learning and higher normalized returns across training steps.

## H.2   SHRINK AND PERTURB

We ran Shrink and Perturb with shrink factors 0.99, 0.9, 0.8, and 0.5 to compare its performance and training behavior directly against our proposed method (Ash & Adams, 2020).

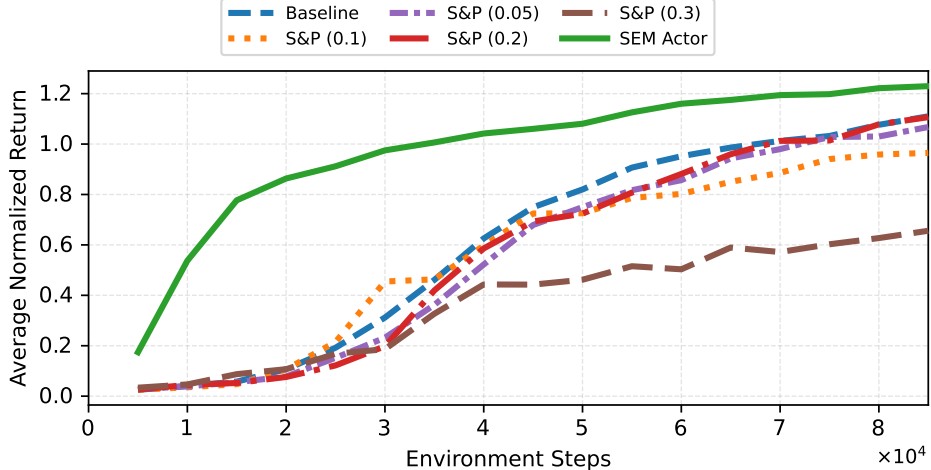

Fig. 16: **Average normalized return on 5 HumanoidBench tasks over 6 seeds.** We compare the Baseline and Shrink-and-Perturb variants with SEM Actor. SEM variation maintains a clear advantage across all S&P levels, achieving faster learning and higher normalized returns.

### H.3 WEIGHT CLIP

We ran Weight Clipping with ranges $(-0.01, 0.01)$, $(-0.001, 0.001)$, and $(-0.1, 0.1)$ to assess how different clipping strengths perform and to compare them directly against our proposed method (Elsayed et al., 2024).

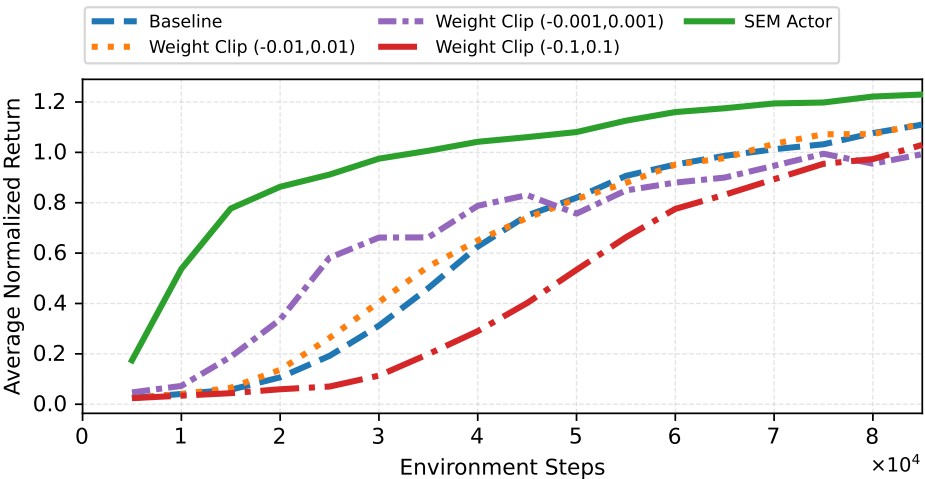

Fig. 17: **Average normalized return on 5 HumanoidBench tasks over 6 seeds.** We compare the Baseline and Weight Clipping variants with SEM Actor. SEM variation achieves faster improvement and higher normalized returns across environment steps.

### H.4 RESET LAYER

We applied the reset layer baseline by reinitializing weights with Xavier initialization, allowing us to evaluate how fully resetting a layer compares to the performance and stability of our proposed method (Nikishin et al., 2022).

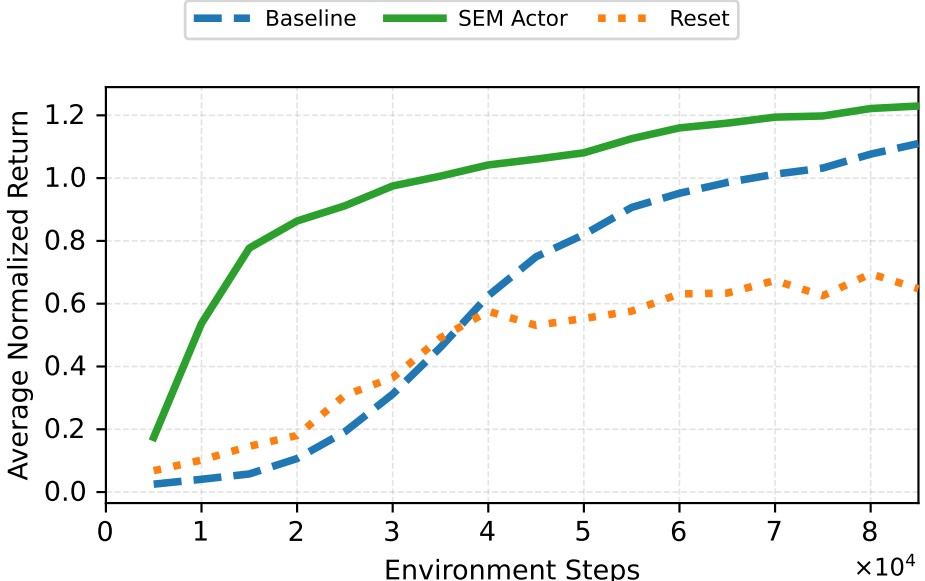

Fig. 18: **Average normalized return on 5 HumanoidBench tasks over 6 seeds.** We compare the Baseline and Reset Layer with SEM Actor. SEM variation accelerates learning and improves normalized return across environment steps, outperforming both the baseline and the reset-layer strategy.

## H.5 SPECTRAL NORM

We applied Spectral Normalization to constrain layer norms and stabilize training, allowing us to directly compare its behavior and performance against the results achieved by our proposed method (Gogianu et al., 2021).

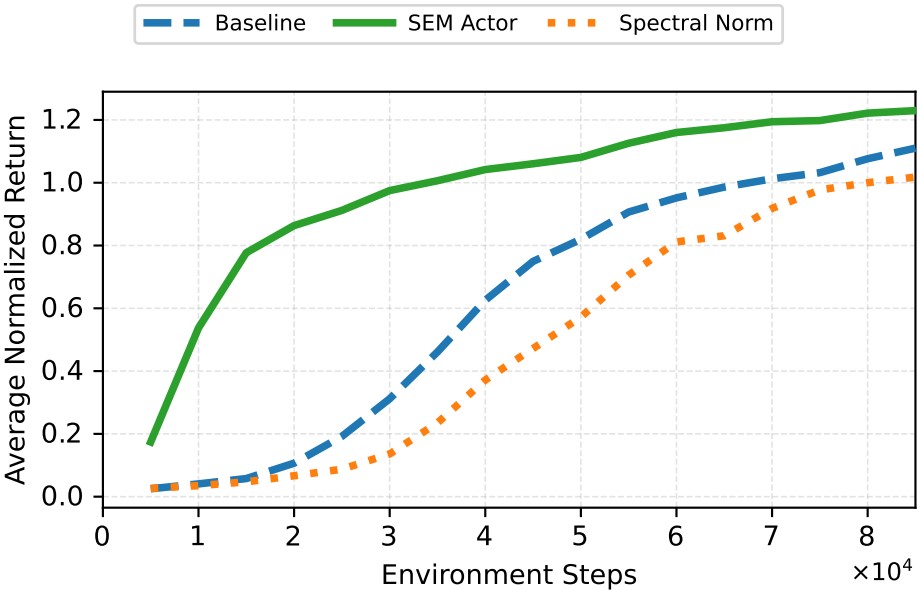

Fig. 19: **Average normalized return on 5 HumanoidBench tasks over 6 seeds.** We compare the Baseline and Spectral Norm with SEM Actor. SEM variation outperforms both the baseline and Spectral Norm across training, achieving faster learning and higher normalized returns.

## H.6 FEATURE NORM

We applied Feature Norm by normalizing activations to unit L2 norm, allowing us to evaluate its effect on representation stability and directly compare its behavior with the performance of our proposed method (Kumar et al., 2023).

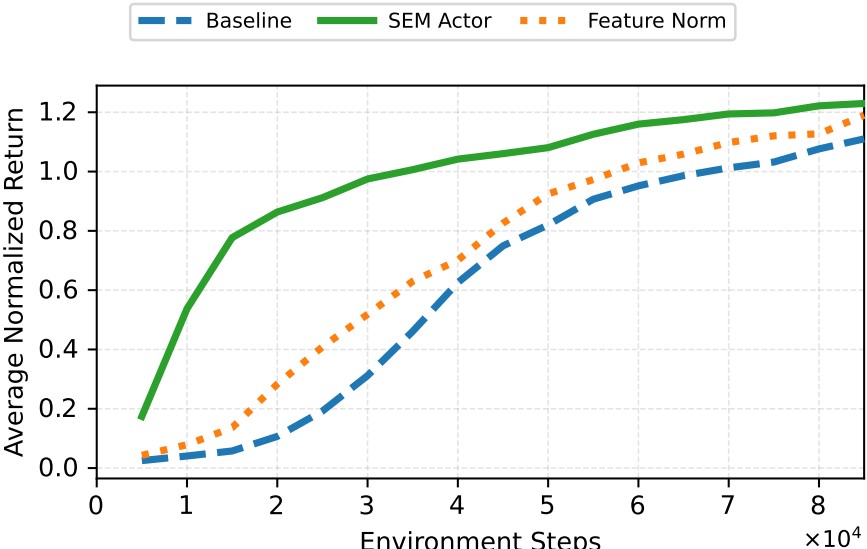

Fig. 20: **Average normalized return on 5 HumanoidBench tasks over 6 seeds.** We compare the Baseline and Feature Norm with SEM Actor. SEM variation outperforms both methods, showing faster learning and higher normalized returns throughout training.

## H.7 GELU

We evaluated the GELU activation to assess its impact on learning dynamics and representation quality, enabling a direct comparison between this widely used baseline and the performance achieved by our proposed method (Hinton et al., 2012).

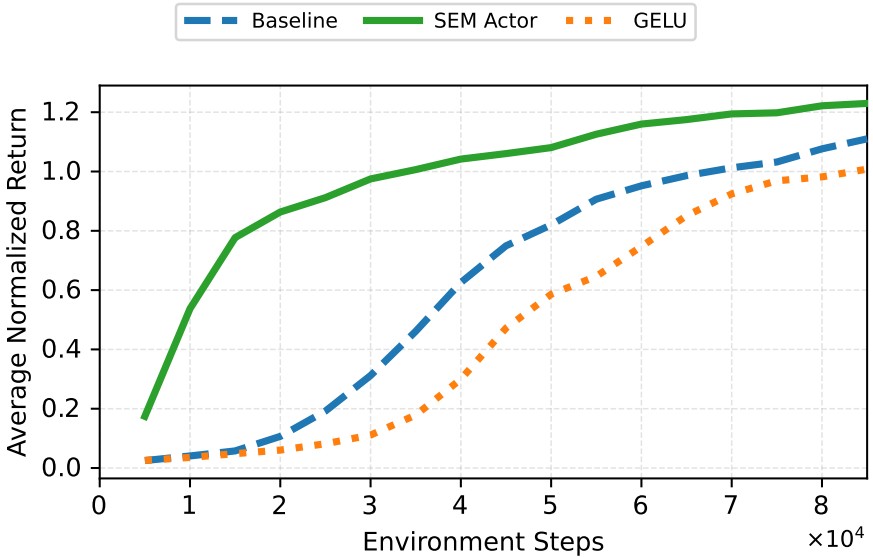

Fig. 21: **Average normalized return on 5 HumanoidBench tasks over 6 seeds.** We compare the Baseline and GELU activation with SEM Actor. SEM variation consistently surpasses both approaches, achieving faster learning and higher normalized returns across training.

## H.8   DROPOUT

We evaluated Dropout using probabilities 0.1, 0.05, and 0.2 to assess its regularization behavior and directly compare its performance and stability against the results achieved by our proposed method (Hendrycks, 2016).

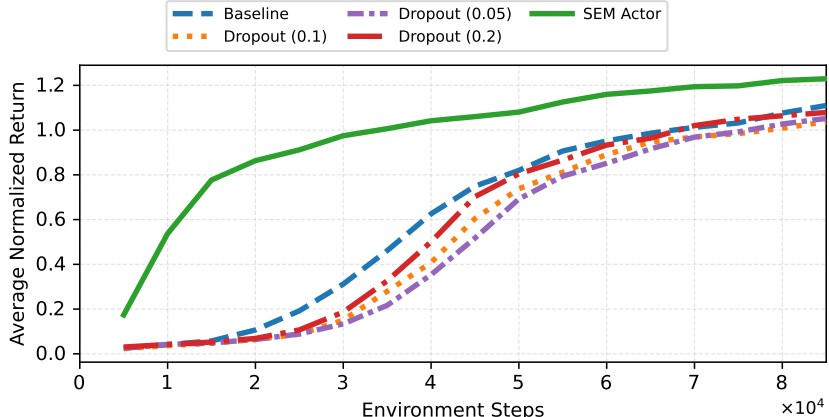

Fig. 22: **Average normalized return on 5 HumanoidBench tasks over 6 seeds.** We compare the Baseline and Dropout variants with SEM Actor. SEM variation consistently shows faster learning progress and achieves higher normalized returns across training.

## H.9   MOES

We evaluate Mixture-of-Experts (MoE) architectures with 1 and 4 experts to study how expert multiplicity influences model capacity, learning dynamics, and overall stability (Ceron et al., 2024c; Willi et al., 2024). These comparisons allow us to assess whether increasing routing flexibility or expert specialization can match the gains provided by SEM. As shown in Fig. 23, both SoftMoE and Top1-MoE track the baseline closely during training, with modest improvements in later stages for the 4-expert configuration. However, none of the MoE variants approach the sample efficiency or final performance achieved by SEM Actor. SEM consistently learns faster and attains higher normalized returns.

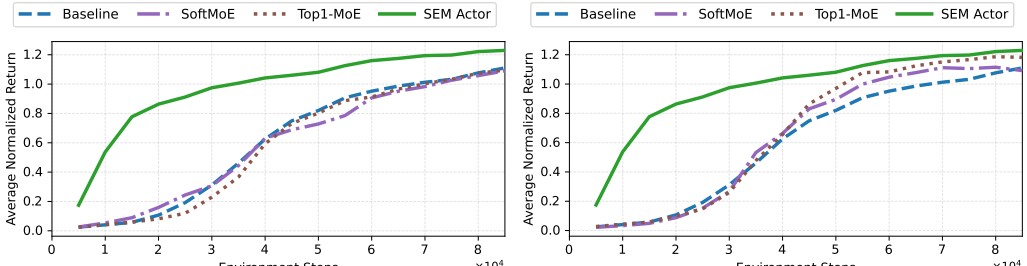

Fig. 23: **Average normalized return on 5 HumanoidBench tasks over 6 seeds.** We compare the Baseline, SoftMoE, Top1-MoE, and SEM Actor. The left plot shows MoE models with 1 expert, and the right plot shows models with 4 experts. MoE variants offer limited gains over the baseline and show slower learning dynamics, while SEM consistently achieves faster progress and higher final returns throughout training.

## H.10   REDO

We evaluated ReDo (Sokar et al., 2023) by varying its dormancy thresholds (0.1, 0.01, 0.001) to study the sensitivity of neuron-reset frequency and to compare its stability and performance to SEM. These thresholds control how aggressively inactive neurons are reset, allowing us to examine whether ReDo's representational refreshing mechanism can match the benefits provided by SEM. As shown in Fig. 24, all ReDo variants track the baseline closely throughout training, with limited

improvements in early or final performance. In contrast, SEM Actor consistently learns faster and achieves higher normalized returns.

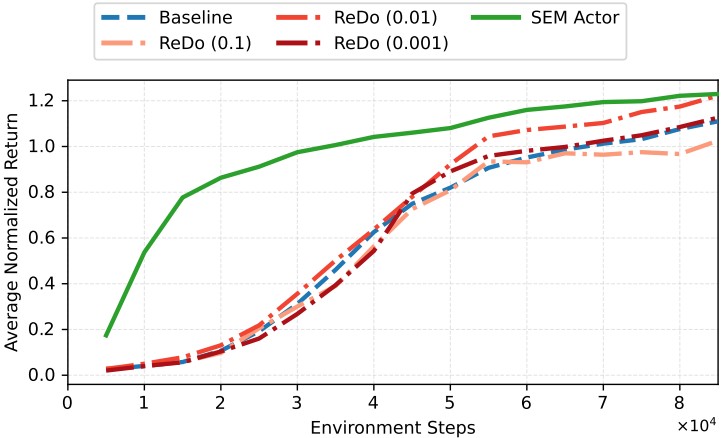

Fig. 24: **Average normalized return on 5 HumanoidBench tasks over 6 seeds.** We compare the Baseline, three ReDo variants (thresholds 0.1, 0.01, 0.001), and SEM Actor. All ReDo configurations remain close to the baseline, showing limited gains across training. SEM achieves faster learning and higher final returns, highlighting the effectiveness of structured embedding modifications over neuron-reset–based approaches.

### H.11 MAGNITUDE PRUNING

We evaluate Magnitude Pruning under two target sparsity levels (50% and 95%) to analyze how pruning severity affects performance and to provide a direct comparison with our proposed method (Graesser et al., 2022; Ceron et al., 2024b). Fig. 25 reports results on 5 HumanoidBench tasks. Overall, both pruning variants lag behind SEM Actor across the entire training horizon. Moderate pruning (50%) retains reasonable learning progress but consistently underperforms SEM in both sample efficiency and final returns. Severe pruning (95%) leads to an early collapse in training, indicating that aggressive weight removal significantly harms representational capacity in this setting. In contrast, SEM maintains stable optimization and achieves substantially higher returns, highlighting the robustness of structured embedding modifications compared to unstructured pruning.

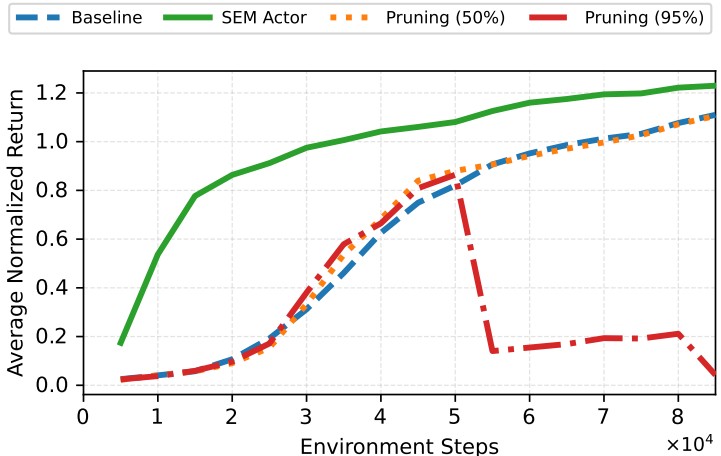

Fig. 25: **Average normalized return on 5 HumanoidBench tasks over 6 seeds.** We compare the Baseline, SEM Actor, and Magnitude Pruning at 50% and 95% target sparsity. SEM consistently learns faster and reaches higher final returns than both pruning variants. Moderate pruning (50%) provides limited gains but remains inferior to SEM, while high-sparsity pruning (95%) collapses training early, demonstrating the instability of aggressive unstructured pruning.

## I  HYPERPARAMETERS

In this section, we list the hyperparameters used across our experimental settings.

Table 2: Wall-clock training time (hh:mm) for the **actor** on the H1-hand humanoid benchmark under default settings. We compare FastTD3 and FastTD3+SEM; lower is better.

| Game | Actor | |
|---|---|---|
| | *FastTD3* | *FastTD3+SEM* |
| h1hand-walk | 2:31 h | 2:42 h |
| h1hand-stand | 2:29 h | 2:20 h |
| h1hand-run | 2:46 h | 2:34 h |
| h1hand-stair | 4:09 h | 4:13 h |
| h1hand-slide | 5:35 h | 5:24 h |

Table 3: Default hyperparameter settings for the PPO agent on Isaac Gym.

| Hyper-parameter | Value |
|---|---|
| Adam's ($\epsilon$) | 1e-5 |
| Adam's learning rate | 2.6e-3 |
| Dense Activation Function | Tanh |
| Dense Width | 256 |
| Discount Factor | 0.99 |
| Number of Dense Layers | 3 |
| Number of environments | 4096 |

Table 4: Default hyperparameter settings for the fastTD3 agent on the humanoid bench.

| Hyper-parameter | Value |
|---|---|
| Critic Hidden Dim | 1024 |
| Actor Hidden Dim | 512 |
| Critic Learning Rate | 3e-4 |
| Actor Learning Rate | 3e-4 |
| Discount Factor | 0.99 |
| Dense Activation Function | ReLU |
| Number of Dense Layers | 4 |
| Number of environments | 128 |
| Number of atoms | 101 |

Table 5: Default hyperparameter settings for the PPO agent on Atari.

| Hyper-parameter | Value |
|---|---|
| Adam's ($\epsilon$) | 1e-5 |
| Adam's learning rate | 2.5e-4 |
| Conv. Activation Function | ReLU |
| Convolutional Width | 32,64,64 |
| Dense Activation Function | ReLU |
| Dense Width | 512 |
| Normalization | None |
| Discount Factor | 0.99 |
| Number of Convolutional Layers | 3 |
| Number of Dense Layers | 2 |
| Reward Clipping | True |
| Weight Decay | 0 |

## J    LEARNING CURVES FOR EACH GAME

To complement the aggregate results reported in the main text (see section 4 and section 5), we provide full learning curves for each environment in the benchmark. These plots illustrate training dynamics across seeds and highlight differences in sample efficiency and stability between +SEM and its corresponding baseline (FastTD3/FastTD3-SimbaV2/FastSAC). The set of robotics tasks follows those used in the FastTD3 benchmark (Seo et al., 2025).

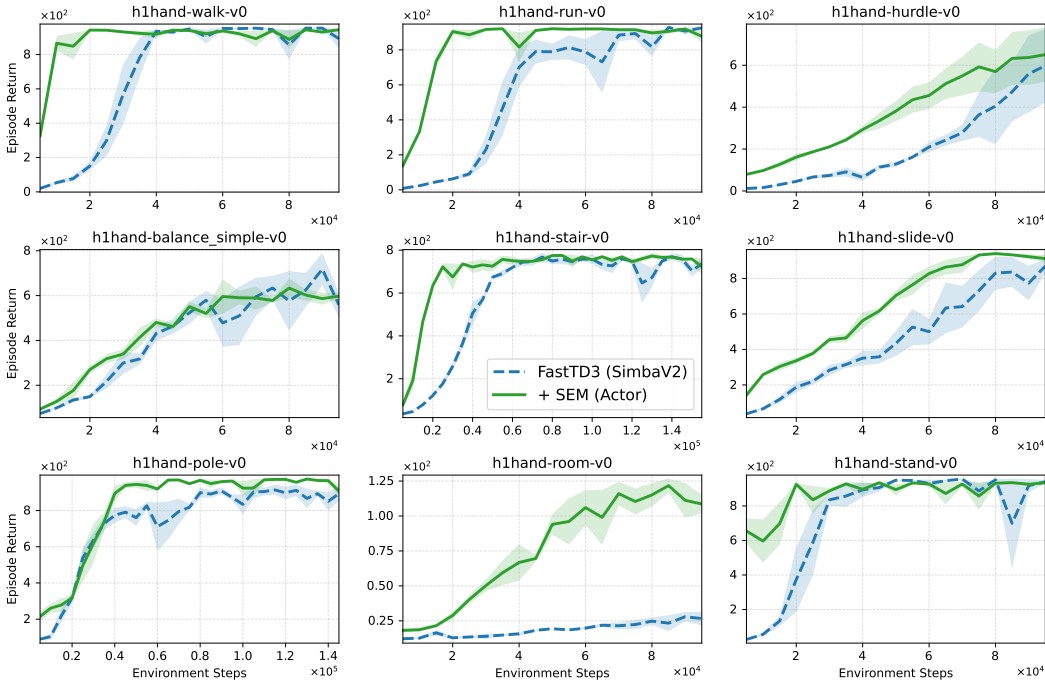

Fig. 26: **Learning curves on 9 `h1hand` tasks. FastTD3+SimbaV2 (blue, - -) vs. + SEM (Actor) (green, —).** Curves show the mean episode return across 6 seeds. Axes are independently scaled per subplot for readability. SEM (Actor) consistently accelerates learning and achieves higher or comparable final returns on most tasks.

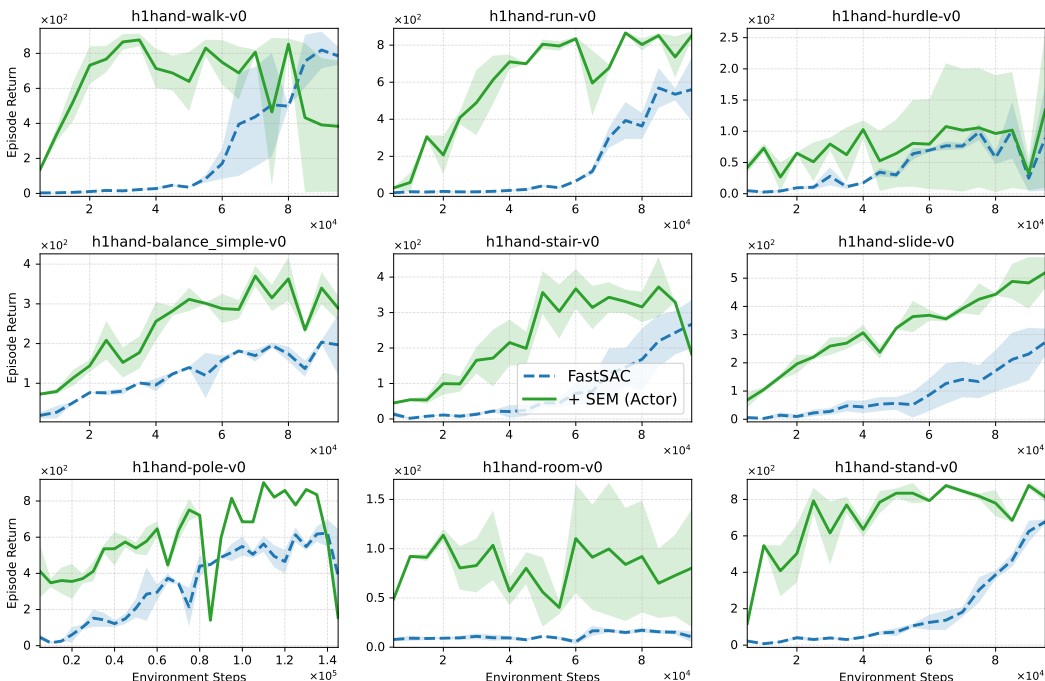

Fig. 27: **Learning curves on 9 `h1hand` tasks. FastSAC (blue, - -) vs. + SEM (Actor) (green, —).** Curves show the mean episode return across 6 seeds. Axes are independently scaled per subplot for readability. SEM (Actor) generally accelerates learning and achieves higher final returns on most tasks.

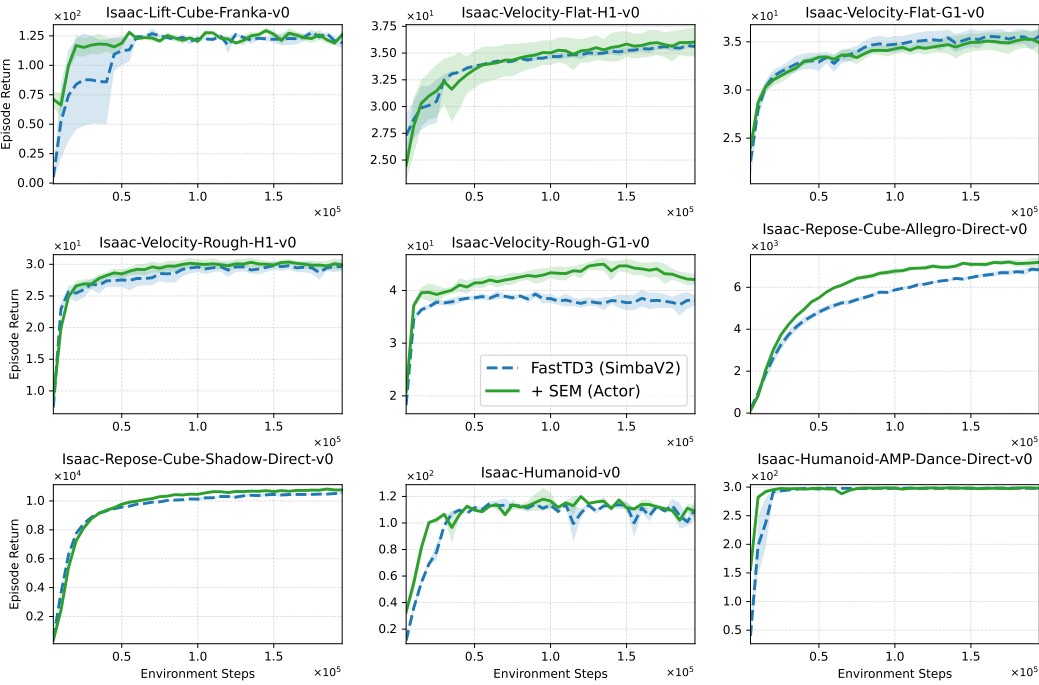

Fig. 28: **Learning curves on 9 IsaacGym tasks. FastSAC (blue, - -) vs. + SEM (Actor) (green, —).** Curves show the mean episode return across 6 seeds. Axes are independently scaled per subplot for readability. SEM (Actor) generally accelerates learning.

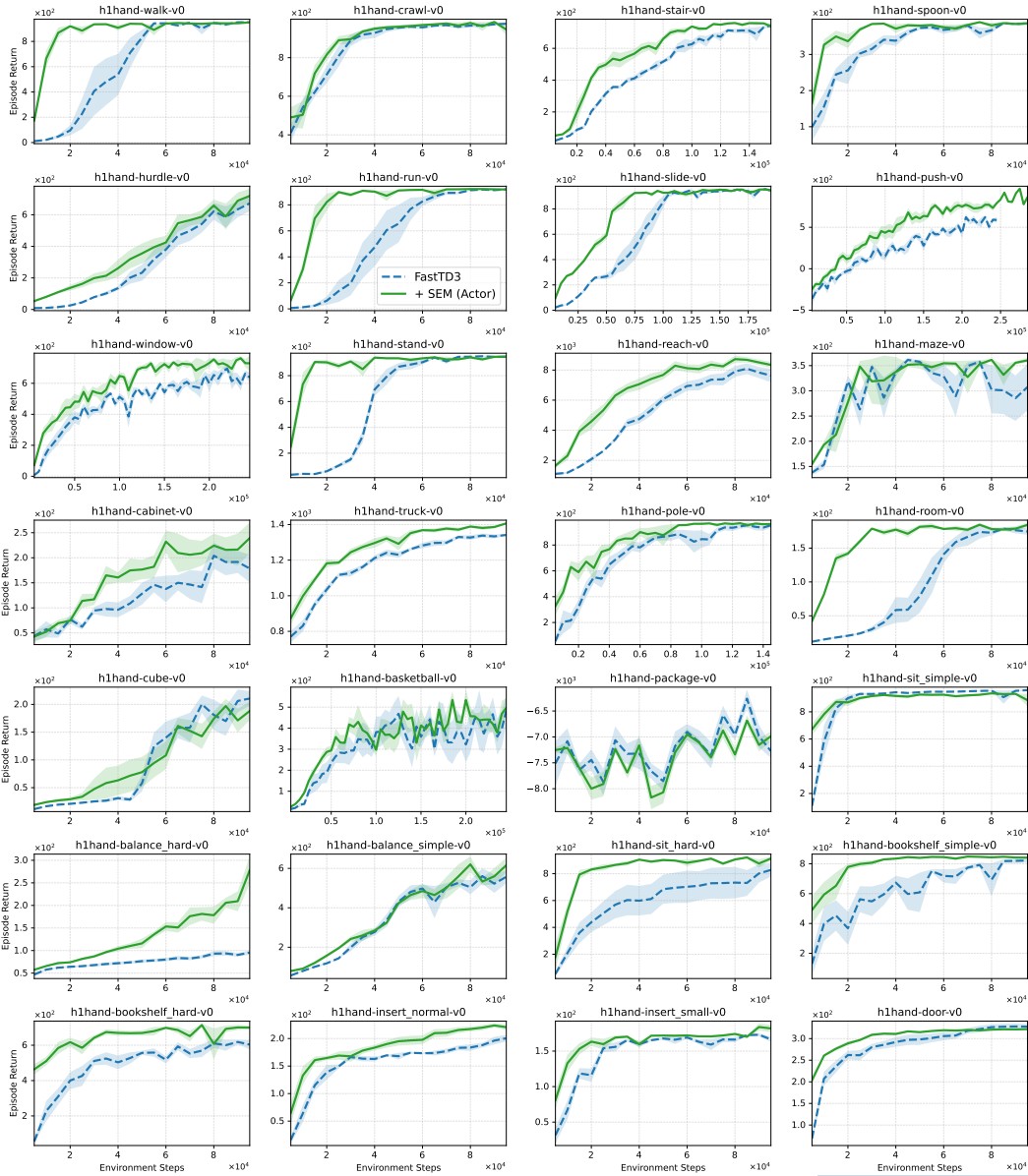

Fig. 29: **Learning curves on 28 `h1hand` tasks** (Sferrazza et al., 2024). **FastTD3 (blue, - -) vs. + SEM (Actor) (green, —).** Curves show the mean episode return across 6 seeds. Axes are independently scaled per subplot for readability. SEM (Actor) typically achieves faster learning and equal or higher final return on most tasks.

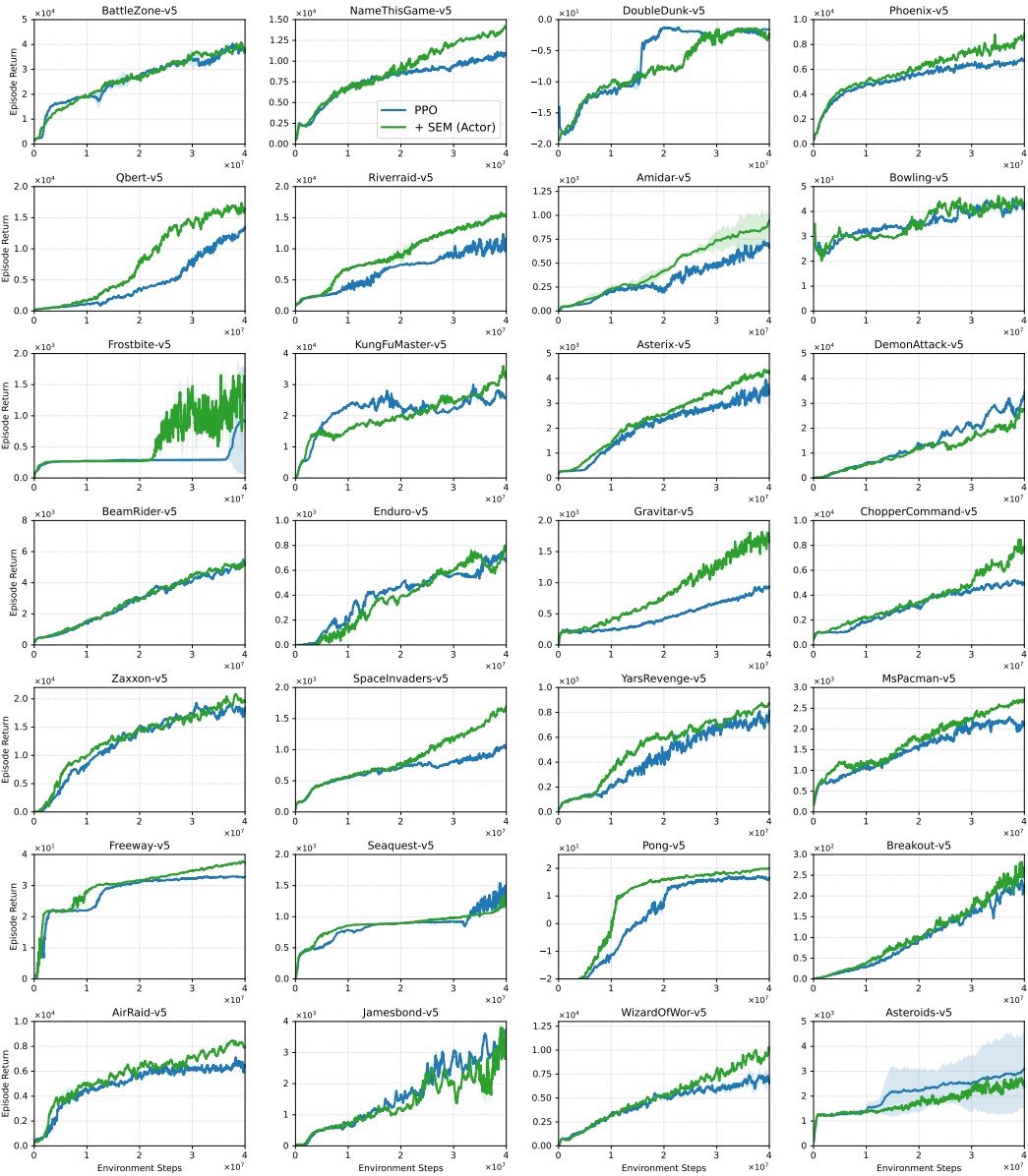

Fig. 30: **Learning curves on Atari game** (Aitchison et al., 2023). **PPO (blue, - -) vs. + SEM (Actor) (green, —).** Curves show the mean episode return across 3 seeds. SEM (Actor) typically achieves faster learning and equal or higher final return on most tasks.

# K ADDITIONAL EXPERIMENTS ON HUMANOIDBENCH

We evaluate +SEM beyond the tasks proposed in FastTD3 by considering additional environments from the Humanoid benchmark (Sferrazza et al., 2024). These experiments assess the scalability of SEM across different robot morphologies and task sets. We include environments featuring the H1 robot without hands and the Unitree G1 with three-finger hands.

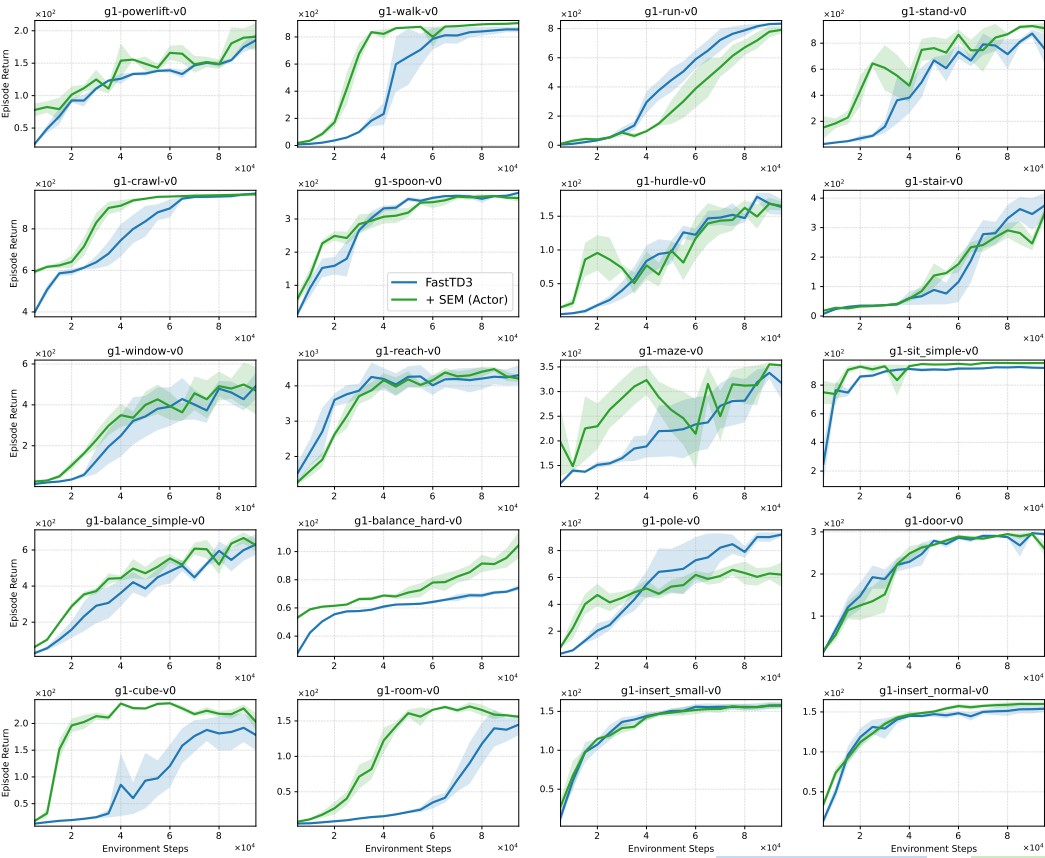

Fig. 31: **Learning curves on 16 h1 tasks (Sferrazza et al., 2024). FastTD3 (blue, - -) vs. + SEM (Actor) (green, —).** Curves show the mean episode return across 6 seeds. Axes are independently scaled per subplot for readability. SEM (Actor) typically achieves faster learning and equal or higher final return on most tasks.

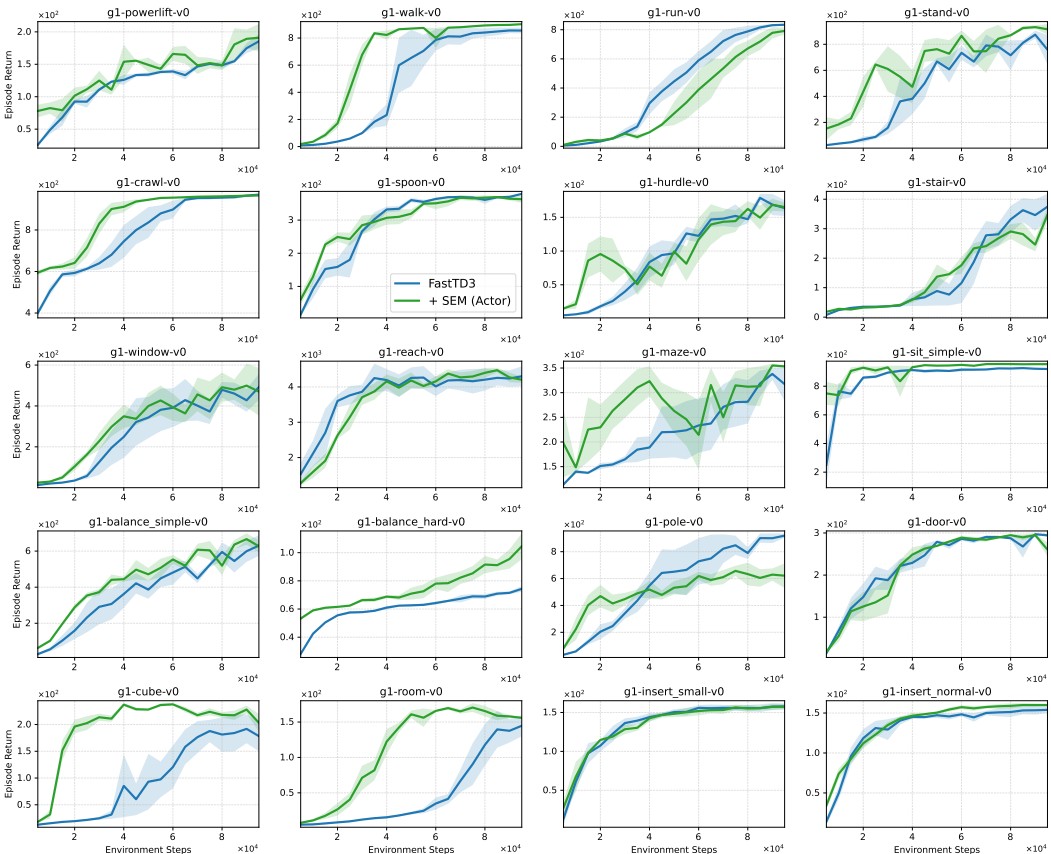

Fig. 32: **Learning curves on 20 g1 tasks** (Sferrazza et al., 2024). **FastTD3 (blue, - -) vs. + SEM (Actor) (green, —).** Curves show the mean episode return across 6 seeds. Axes are independently scaled per subplot for readability. SEM (Actor) typically achieves faster learning and equal or higher final return on most tasks.

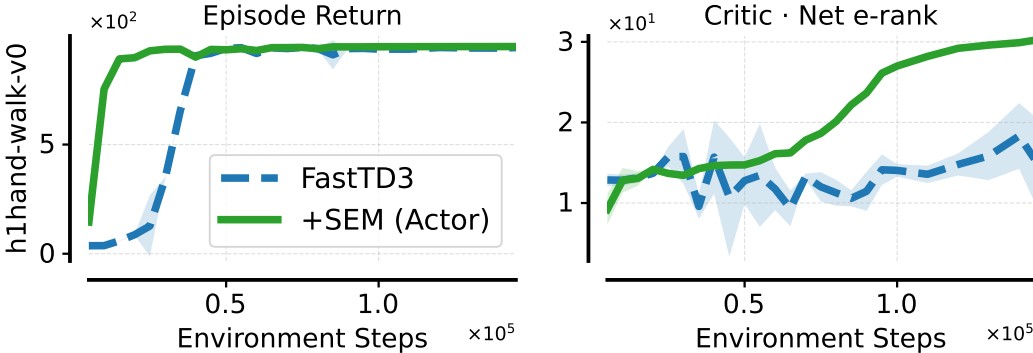

Fig. 33: **Learning and representation diagnostic on `h1hand-walk-v0 task`.** SEM reaches high return earlier and critic effective rank.

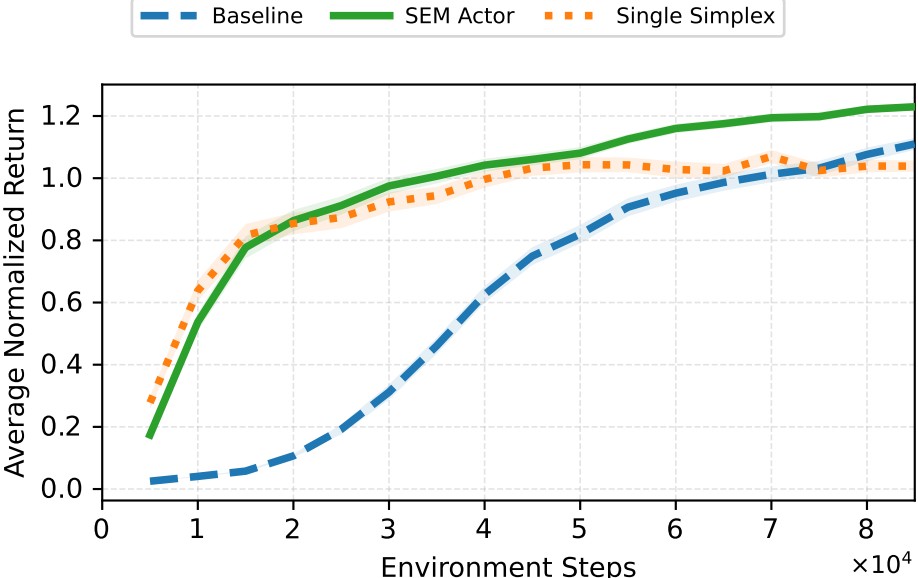

Fig. 34: **Average normalized return on** 5 **HumanoidBench tasks over** 6 **seeds.** We compare the baseline agent (**blue**, − −) with SEM variants applied to the actor. SEM variants accelerate early learning, though a single-simplex configuration tends to plateau.

## L    EVALUATING LAYER-WISE INTERVENTIONS

Selecting which layer to modify is non-trivial, as the search space grows with model depth and architectural complexity. Throughout the paper, we apply SEM to the penultimate layer, following prior work (Gogianu et al., 2021; Kumar et al., 2023; Ceron et al., 2024c; Sokar & Castro, 2025). These studies highlight that the penultimate layer plays a key role in shaping and constraining learned representations in deep RL.

Here, we extend this analysis by evaluating the effect of applying SEM to individual actor layers as well as to all layers simultaneously. Following the experimental setup in section 4, we report results in Fig. 35 and Fig. 36. Across both settings, we observe that applying SEM to deeper layers or to the entire network leads to faster learning and higher final returns compared to intervening on early layers. Notably, applying SEM solely to Layer-3 achieves performance comparable to the full-layer intervention, indicating that modifying a single well-chosen layer is sufficient to capture most of SEM's benefits.

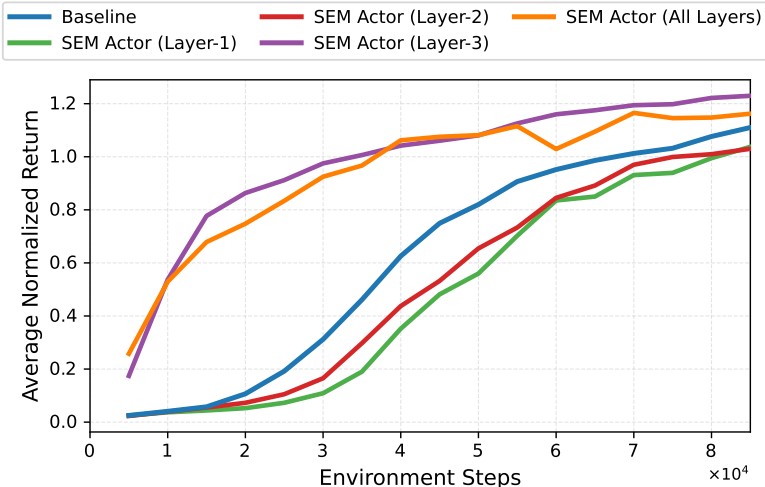

Fig. 35: **Average normalized return on** 5 **HumanoidBench tasks over** 6 **seeds.** We evaluate the effect of applying SEM to individual actor layers (Layer-1, Layer-2, Layer-3) and to all layers with $dim = 64$.

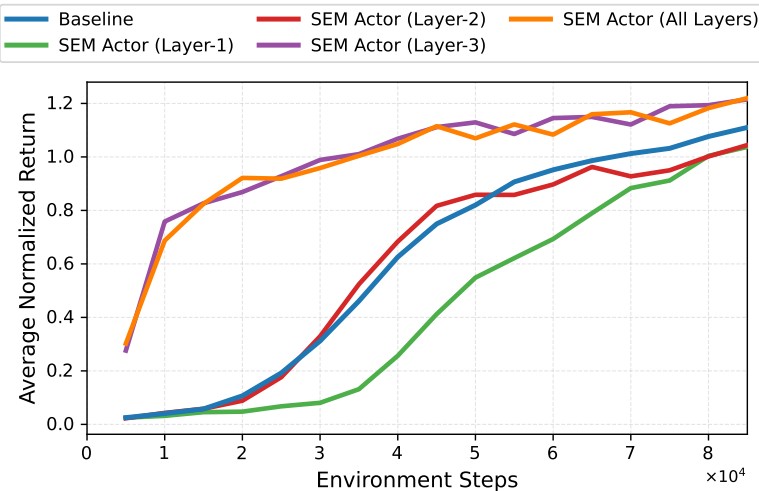

Fig. 36: **Average normalized return on** 5 **HumanoidBench tasks over** 6 **seeds.** We evaluate the effect of applying SEM to individual actor layers (Layer-1, Layer-2, Layer-3) and to all layers with $dim = 128$.

# M   SEM ON VALUE-BASED DEEP RL

We additionally evaluate SEM in value-based deep RL settings. Specifically, we run PQN (Gallici et al., 2025) on 28 Atari games following the experimental setup from section 5 using 3 seeds and training for $40M$ environment steps. As shown in Fig. 38, SEM does not yield consistent improvements over the baseline when averaging performance across all games. However, examining per-game learning curves (see Fig. 37) reveals that SEM provides meaningful gains in both sample efficiency and final performance on several titles (e.g., Asterix, BeamRider). These results suggest that the effectiveness of SEM in value-based methods may depend on game-specific dynamics and representation structure, raising interesting research questions that we leave for future investigation.

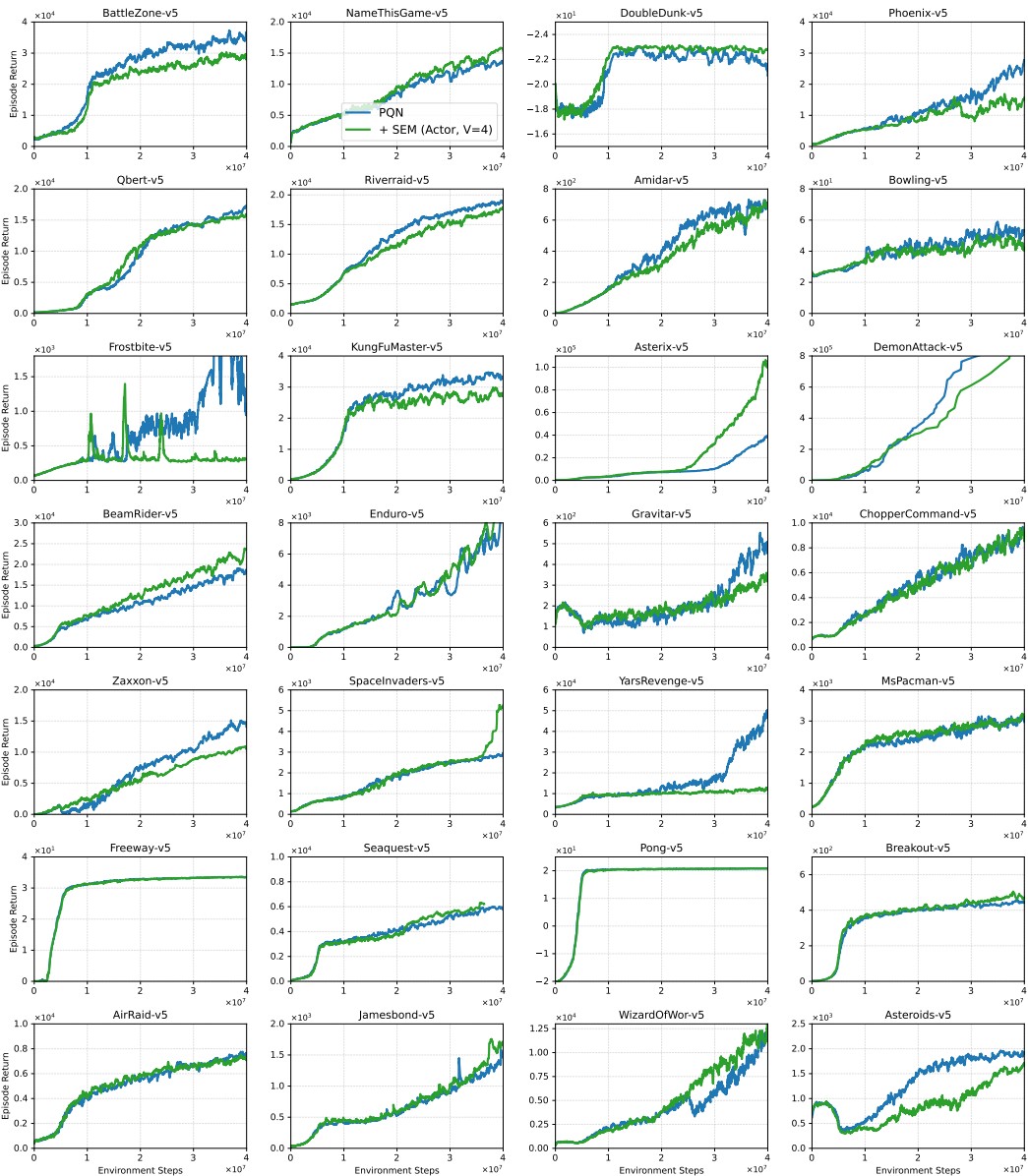

Fig. 37: **Learning curves on** 28 **Atari games (Aitchison et al., 2023). PQN (blue, - -) (Lavoie et al., 2023) vs. + SEM (Actor) (green, —).** Curves show the mean episode return across 3 seeds.

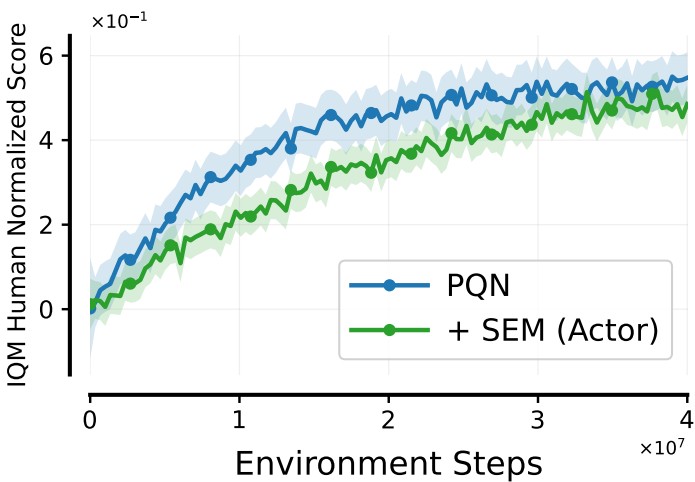

Fig. 38: **IQM scores computed over 40M environment steps over** 18 **games, with** 3 **independent runs each**, and error bars showing 95% stratified bootstrap confidence intervals. PQN (**blue**, - -) (Lavoie et al., 2023) vs. + SEM (Actor) (**green**, —).

