# OpenReview forum: "Simplicial Embeddings Improve Sample Efficiency in Actor–Critic Agents"
_ICLR.cc/2026/Conference — ICLR 2026 Poster_

### Official Review · Reviewer_d8VX · 2025-10-21

**Soundness:** 3
**Presentation:** 4
**Contribution:** 3
**Rating:** 4
**Confidence:** 4

**Summary:**

This paper proposes the use of simplicial embeddings (which lead to sparse and discrete features) as a way to improve the sample efficiency, stability and performance of Reinforcement Learning agents. Furthermore, the paper adds to the body of work showing that non-stationary targets can cause a loss of plasticity, and demonstrates that SEM can mitigate this effect.

**Strengths:**

This paper presents a simple, practical idea as a quick and effective way to improve sample efficiency in Reinforcement Learning, which can be easily used in almost any algorithm. The analysis on why SEM improves performance is deep and uses a variety of metrics previously proposed in the literature. Furthermore, the evaluation is quite thorough, including different algorithms and environments.

**Weaknesses:**

My main problem with this paper is the lack of comparison to prior methods that solve very similar problems. This paper correctly cites a wide range of prior work that propose solutions to the problem of degradation of learned representations under non-stationarity ([1, 2, 3] and many more), causing plasticity loss and reduced performance. Despite this large variety of previous similar work, the paper also exclusively compares against baselines with no previous similar solutions. While many figures show improvement, none show whether this is actually an improvement over prior existing work. For this work to be accepted at a venue of this standard, I would expect that Figures 1, 9 and 10 would include a comparison against prior strategies.

Figures 9 and 10 have inconsistent axes. The left and middle plots present very similar data (million of frames on the x axis, and performance on the y axis), yet are presented under different scales. This inconsistency makes the results a little harder to read.

No source code was included in the supplementary material, and there is no mention of source code being released after review period. This hurts the reproducibility of this work.

Figure 10 uses the Atari-10 benchmark [4], but presents results as IQM performance. The purpose of the Atari-10 paper was to predict the **median** performance, using a specific regression procedure provided by the authors (which weights the importance of each game). Given that the paper reports IQM performance, it appears the procedure was not followed.

There is a typo on line 215.

[1] Lyle, Clare, et al. "Disentangling the Causes of Plasticity Loss in Neural Networks." Conference on Lifelong Learning Agents. PMLR, 2025.

[2] Ceron, Johan Samir Obando, Aaron Courville, and Pablo Samuel Castro. "In value-based deep reinforcement learning, a pruned network is a good network." International Conference on Machine Learning. PMLR, 2024.

[3] Willi, Timon, et al. "Mixture of Experts in a Mixture of RL settings." Reinforcement Learning Conference.

[4] Aitchison, Matthew, Penny Sweetser, and Marcus Hutter. "Atari-5: Distilling the arcade learning environment down to five games." International Conference on Machine Learning. PMLR, 2023.

**Questions:**

How does this method compare against prior plasticity loss mitigation strategies? Does this method provide better performance? Is it better at preventing representation collapse?

It is unclear to me why this paper focuses on actor-critic algorithms specifically. Line 323 provides literature suggesting that non-actor-critic approaches can also benefit from sparsity in representations. Given this, it seems odd to me that the paper is so heavily tied to actor-critic methods (including the title). Why is this the case?

What do the shaded areas on the different figures represent (Standard error, 95% Confidence intervals, etc)? Additionally, how many seeds does Figure 1 use?

I think this paper is valuable to the RL community; however, given the lack of comparison against prior methods, I cannot yet advocate for acceptance. If this issue is remedied, and the method demonstrates superior properties to prior methods, I am willing to raise my score.

---

> ### Author Response · Authors · 2025-11-20
> **Rebuttal 1/3**
>
> We sincerely appreciate the reviewer’s valuable comments. We are happy that the reviewer found that “the paper has an excellent presentation”. We're also pleased that the reviewer recognized “SEM as a practical idea as a quick and effective way to improve sample efficiency in Reinforcement Learning, which can be easily used in almost any algorithm”; that the “analysis on why SEM improves performance is deep and uses a variety of metrics previously proposed in the literature”, and that “the evaluation is quite thorough, including different algorithms and environments.”
>
> We also value the reviewer’s view that this work is “valuable for the RL community.” Below we address the remaining concerns.
>
> > For this work to be accepted at a venue of this standard, I would expect that Figures 1, 9 and 10 would include a comparison against prior strategies.
>
> Thank you for pointing this out. We agree that including comparisons to prior plasticity-related strategies is beneficial, and we incorporate the most relevant baselines [1–12] where applicable. Running all experiments for the three figures (1, 9, and 10) as suggested is unfortunately infeasible during the rebuttal period, as it would require substantial computational resources (6 RL algorithms × 50 games × 6 seeds × 8 methods, totaling approximately 14,000 runs).
>
> That said, we consider this an important suggestion. Following the setup in the section **Comparing SEM to Other Regularization Methods**, we evaluate additional interventions commonly used to mitigate plasticity loss and optimization issues in deep RL. Specifically, we assess 8 extra methods such as Reset [1], L2 [2], Regen [3], Dropout[4], Shrink-and-Perturb [5], GELU activation function [6], Weight Clipping [7], and Spectral Norm [8], each with 6 seeds across 5 humanoid benchmarks. These experiments are additional to the ones presented in Fig. 7, which includes other relevant methods like, Gumbel + straight-through [9, 10], Vector Quantization [11], and C-RELU [12].
>
> Please see Appendix H (Additional Baselines) for the average normalized return across tasks for each intervention. From these results, we observe that all methods outperform SEM. We also evaluate a range of hyperparameter values where needed to ensure fairness and verify optimality; these results are likewise provided here [https://bit.ly/44kh4aQ], folder: images/extra_baselines].
>
> We would like to emphasize that the goal of the paper is not to claim that SEM is universally superior to all techniques. Rather, our contribution is to show that SEM provides a lightweight and effective mechanism for mitigating representation collapse in deep RL.
>
> In contrast to methods like MoEs [13], ReDo [14], or pruning-based interventions [15], SEM does not require exhaustive hyperparameter searches (e.g., which layer to reset, when to intervene) nor the additional implementation complexity those methods demand. We exclude these methods because they require substantial tuning and architectural modifications that fall outside the scope of our lightweight design objective.
>
> [1] Nikishin, et al. " The Primacy Bias in Deep Reinforcement Learning”, ICM’22
>
> [2] Kumar, et al. "Offline Q-Learning on Diverse Multi-Task Data Both Scales And Generalizes”, ICLR’23
>
> [3] Kumar, et al. “Maintaining Plasticity in Continual Learning via Regenerative Regularization.”, CoLLAs’24
>
> [4] Hendrycks, et al. “Gaussian Error Linear Units (Gelus)." 2016.
>
> [5] T. Ash, et al. “On Warm-Starting Neural Network Training”. NeurIPS’20
>
> [6]  Hinton , et al. “Improving neural networks by preventing co-adaptation of feature detectors, JMLR‘12
>
> [7] Elsayed, et al. "Weight Clipping for Deep Continual and Reinforcement Learning.”, RLC’24
>
> [8] Gogianu, et al. "Spectral Normalisation for Deep Reinforcement Learning: An Optimisation Perspective” ICLR’21
>
> [9] Jang,, et al. "Categorical reparameterization with gumbel-softmax”, 2017.
>
> [10] Maddison, et al. " The concrete distribution: A continuous relaxation of discrete random variables”, 2017.
>
> [11] van den Oord, et al. "Neural discrete representation learning”, 2018
>
> [12] Abbas, et al. "Loss of Plasticity in Continual Deep Reinforcement Learning”. CoLLAs’23.
>
> [13] Willi, Timon, et al. "Mixture of Experts in a Mixture of RL settings.” RLC’24
>
> [14]  Sokar et al. "The Dormant Neuron Phenomenon in Deep Reinforcement Learning", ICML’23
>
> [15] Ceron et al. "In value-based deep reinforcement learning, a pruned network is a good network." ICML’24.

---

> > ### Author Response · Authors · 2025-11-20
> > **Rebuttal 2/3**
> >
> > > Figures 9 and 10 have inconsistent axes. The left and middle plots present very similar data (million of frames on the x axis, and performance on the y axis), yet are presented under different scales. This inconsistency makes the results a little harder to read.
> >
> > Thanks for the suggestion. We have unified the axis scaling across the corresponding plots to improve readability and avoid unnecessary visual discrepancies.
> >
> > > No source code was included in the supplementary material, and there is no mention of source code being released after the review period. This hurts the reproducibility of this work.
> >
> > We have attached the full implementation for FastTD3, and we will be adding integration into the FastTD3 repository to ensure easy adoption and benchmarking by the community [source code: https://bit.ly/44kh4aQ], folder: code].
> >
> > > Figure 10 uses the Atari-10 benchmark [4], but presents results as IQM performance. The purpose of the Atari-10 paper was to predict the median performance, using a specific regression procedure provided by the authors (which weights the importance of each game). Given that the paper reports IQM performance, it appears the procedure was not followed.
> >
> > Thanks for bringing this to our attention. Our Atari evaluation in Fig. 10 uses the standard Atari-10 suite [1], supplemented with additional games from the Atari-20 subset [2], in order to evaluate the generality of our method across a larger set of environments (see all learning curves in Fig. 18). In total, we evaluate 28 games and report IQM returns (Fig. 10) following the protocol of [3,4,5,6].
> >
> > We added clarity when referring to the Atari game setup and included the missing citation to avoid any confusion.
> >
> > [1]. Aitchison et all, "Atari-5: Distilling the arcade learning environment down to five games." International Conference on Machine Learning. PMLR, 2023.
> >
> > [2]. Fedus et all “Revisiting Fundamentals of Experience Replay”, ICML’20.
> >
> > [3]. Agarwal et all “Deep Reinforcement Learning at the Edge of the Statistical Precipice”, NeurIPS’21.
> >
> > [4]. Obando-Ceron et all “Mixtures of Experts Unlock Parameter Scaling for Deep RL”, ICML’24.
> >
> > [5]. Obando-Ceron et all “In value-based deep reinforcement learning, a pruned network is a good network”, ICML’24.
> >
> > [6]. Sokar et all “Mind the GAP! The Challenges of Scale in Pixel-based Deep Reinforcement Learning”, NeurIPS 2025
> >
> > > There is a typo on line 215.
> >
> > Thanks for the catch. We fixed this typo as follows: “We report aggregate performance across the five tasks and six seeds, with full details provided in the App. E.”

---

> > > ### Author Response · Authors · 2025-11-20
> > > **Rebuttal 3/3**
> > >
> > > # Questions
> > >
> > > > It is unclear to me why this paper focuses on actor-critic algorithms specifically. Line 323 provides literature suggesting that non-actor-critic approaches can also benefit from sparsity in representations. Given this, it seems odd to me that the paper is so heavily tied to actor-critic methods (including the title). Why is this the case?
> > >
> > > SEM is general and can be applied to non–actor–critic methods as well. Our choice to focus on actor–critic settings was deliberate:
> > >
> > > 1. These methods are particularly sensitive to representation instability due to their bootstrapped value updates.
> > > 1. They cover a broad and widely used family of deep RL algorithms (TD3, SAC, PPO, etc.).
> > > 1. Current actor–critic algorithms remain far from sample-efficient, despite the substantial gains in wall-clock training speed achieved in recent years [1]. Ideally we would like to improve in both axes.
> > > 1. By focusing on this family of models, we were able to perform a more thorough and mechanistic analysis.
> > >
> > > Extending SEM to non–actor–critic algorithms is promising future work, but would require substantial computational resources to investigate thoroughly. We included a discussion of this limitation and related future directions in the first submission, and we can expand it further if required.
> > >
> > > [1]. Seo et al. "FastTD3: Simple, Fast, and Capable Reinforcement Learning for Humanoid Control”, 2025
> > >
> > > > What do the shaded areas on the different figures represent (Standard error, 95% Confidence intervals, etc)? Additionally, how many seeds does Figure 1 use?
> > >
> > > The shaded regions represent 95% stratified bootstrap confidence intervals, following common practice [1,2]. We have incorporated this information into the paper to improve clarity, and the updated text is shown in blue in lines 215/411.
> > >
> > > Figure 1 reports results over 3 independent seeds, consistent with the setup in Sokar et al. (ICML ’23) [1]. Similarly, we have added this information to the caption of Fig.1, and the updated text is shown in blue in line 175 .
> > >
> > > [1.] Sokar et al. "The Dormant Neuron Phenomenon in Deep Reinforcement Learning", ICML’23
> > >
> > > [2] Agarwal et all “Deep Reinforcement Learning at the Edge of the Statistical Precipice”, NeurIPS’21.
> > >
> > > > I think this paper is valuable to the RL community; however, given the lack of comparison against prior methods, I cannot yet advocate for acceptance. If this issue is remedied, and the method demonstrates superior properties to prior methods, I am willing to raise my score.
> > >
> > > We appreciate the reviewer’s willingness to raise their score contingent on comparisons to prior methods. We ran these comparisons and added the clarifications above into the revised version. Please let us know if you still feel there is something missing.

---

> > > > ### Comment · Reviewer_d8VX · 2025-11-23
> > > >
> > > > Thank you for your response, answering many of my questions and addressing many concerns.
> > > >
> > > > > For this work to be accepted at a venue of this standard, I would expect that Figures 1, 9 and 10 would include a comparison against prior strategies.
> > > >
> > > > I appreciate the additional experiments, as I believe they will strengthen the paper. I do still think the paper would be improved by the inclusion of more modern techniques (specifically MoEs [13], ReDo [14], or pruning-based interventions [15] as you mention), since many of the provided techniques are rather outdated in this specific area of work and provide little value.
> > > >
> > > > > We would like to emphasize that the goal of the paper is not to claim that SEM is universally superior to all techniques. Rather, our contribution is to show that SEM provides a lightweight and effective mechanism for mitigating representation collapse in deep RL.
> > > >
> > > > I think this is a fair point which I will take into consideration.
> > > >
> > > > > It is unclear to me why this paper focuses on actor-critic algorithms specifically. Line 323 provides literature suggesting that non-actor-critic approaches can also benefit from sparsity in representations. Given this, it seems odd to me that the paper is so heavily tied to actor-critic methods (including the title). Why is this the case?
> > > >
> > > > I find all of the given arguments for this question to be weak/unconvincing.
> > > >
> > > > > These methods are particularly sensitive to representation instability due to their bootstrapped value updates.
> > > >
> > > > Do you have any evidence to suggest they are more sensitive than methods such as value-based methods?
> > > >
> > > > > They cover a broad and widely used family of deep RL algorithms (TD3, SAC, PPO, etc.).
> > > >
> > > > While true, this isn't a reason to tie the paper to actor-critic methods.
> > > >
> > > > > Current actor–critic algorithms remain far from sample-efficient, despite the substantial gains in wall-clock training speed achieved in recent years [1]. Ideally we would like to improve in both axes.
> > > >
> > > > Again, while true, sample efficiency is still extremely valuable in other areas like value-based RL.
> > > >
> > > > > By focusing on this family of models, we were able to perform a more thorough and mechanistic analysis.
> > > >
> > > > I think this points to the reason for not including other methods to be "they're faster to run", rather any substantial reason.
> > > >
> > > > In my opinion, the tie to actor-critic methods is still unnecessary. I'd prefer the authors to simply state "we leave the application of Simplicial Embedding in other areas such as value-based methods to future work", rather than making the method sound specific to this class of algorithms.
> > > >
> > > > While I still have many criticisms of this paper and wouldn't mind if it was rejected and improved for a different venue, I still lean slightly more towards accepting than rejecting and therefore will raise my score to a 6.

---

> > > > > ### Author Response · Authors · 2025-11-28
> > > > >
> > > > > Thank you again for the thoughtful and constructive follow-up. Your comments substantially shaped the latest revision of the paper, and we genuinely appreciate the time you invested in helping us strengthen the work. Below we addressed your two major concerns.
> > > > >
> > > > > > I appreciate the additional experiments, as I believe they will strengthen the paper. I do still think the paper would be improved by the inclusion of more modern techniques (specifically MoEs [13], ReDo [14], or pruning-based interventions [15] as you mention), since many of the provided techniques are rather outdated in this specific area of work and provide little value.
> > > > >
> > > > > Thank you for the suggestion. We agree that evaluating SEM against more recent intervention mechanisms is important for contextualizing the paper contribution. In the updated version, we have incorporated all three families of methods highlighted by the reviewer:
> > > > > - Mixture-of-Experts (SoftMoE, Top-1 MoE, 1-expert and 4-expert variants)
> > > > > - ReDo (dormant-neuron reset, evaluating three threshold values [0.1, 0.01, 0.001])
> > > > > - Magnitude Pruning (50% and 95% sparsity)
> > > > >
> > > > > These new results are reported in Appendix Sections H.10–H.12. Across all settings, the conclusions remain consistent. SEM remains more stable and yields higher returns. We believe these additions meaningfully strengthen the empirical evaluation and directly address the reviewer’s concern.
> > > > >
> > > > > > In my opinion, the tie to actor-critic methods is still unnecessary.
> > > > >
> > > > > We appreciate the reviewer’s perspective. Our intention is not to suggest that SEM is restricted to actor–critic algorithms. To clarify this, we have added explicit discussion emphasizing that SEM operates on learned representations and can be applied to other deep RL methods (see section 5, Empirical  Evaluation).
> > > > >
> > > > > To support this point empirically, we now include additional experiments applying SEM to value-based RL. In particular, we evaluate SEM within PQN on 28 Atari games using 3 seeds (Appendix Section M). While the improvements are more modest and game-dependent, SEM provides measurable gains in some  games. These results highlight that SEM can be used beyond actor–critic settings, while also motivating a more systematic investigation into when geometric constraints most effectively benefit value-based methods. Corresponding updates have been made to the Limitations and Future Work sections.
> > > > >
> > > > > -----
> > > > >
> > > > > We believe we have adequately addressed the concerns you raised. If you have any additional thoughts or questions regarding our responses, we would be more than happy to clarify them. If no concerns remain, we kindly invite you to consider revising your score, as you initially noted that “this paper is valuable to the RL community; however, given the lack of comparison against prior methods, I cannot yet advocate for acceptance.”
> > > > >
> > > > > Thank you very much for all your suggestions!

---

### Official Review · Reviewer_A2Cz · 2025-10-27

**Soundness:** 3
**Presentation:** 3
**Contribution:** 3
**Rating:** 6
**Confidence:** 3

**Summary:**

The paper studies representation collapse as a root cause of poor sample efficiency and training instability in deep RL, arguing that non-stationary, bootstrapped targets make critics ill-conditioned and prone to drifting value estimates. It proposes Simplicial Embeddings (SEM)—a lightweight, group-wise softmax projection that constrains latent features to a product of simplices—to bound features, induce sparsity, and maintain diversity, thereby stabilizing learning. The proposed method is proven to be effective on different base RL algorithms across mainstream benchmarks.

**Strengths:**

**Clear problem framing.** Ties critic instability to non-stationarity and representation collapse with an accessible toy study.

**Simple yet effective, plug-and-play mechanism:** SEM is like an activation with better gradient properties; no extra losses or estimators.

**Broad robustness:** Benefits persist across data-limited settings and simplified agent variants; compares against reasonable discrete/structured alternatives.

**Weaknesses:**

**Inappropriate toy examples.** The CIFAR-10 supervised learning experiment with shuffled labels is not a straightforward toy example for demonstrating the non-stationarities in value learning.

**L, V selection.** It is weird that the limited SEM capacity leads to better performance. Especially the setting that $L=1, V=64$ almost achieves the best performance in Fig. 7 if the y-axes are shared. Doesn't that mean the softmax operator is the best regularizer?

**Questions:**

1. What's the best setting of L and V? Did you change the values of L and V depending on the benchmark and task?

---

> ### Author Response · Authors · 2025-11-20
> **Rebuttal 1/2**
>
> We thank the reviewer for their feedback! We are happy that the reviewer found the “problem framing clear”, appreciated the method as a “simple yet effective, plug-and-play mechanism” and recognized its “broad robustness” across architectures and environments.
>
> > Inappropriate toy examples. The CIFAR-10 supervised learning experiment with shuffled labels is not a straightforward toy example for demonstrating the non-stationarities in value learning.
>
> Deep RL is characterized by non-stationary, self-generated targets arising from bootstrapped updates, a combination that frequently leads to feature drift, dormant units, and representation collapse. To diagnose this, a growing body of RL research has adopted shuffled-label CIFAR-10, as we do, as a controlled proxy for inducing non-stationary targets and studying these instabilities [1–7]:
>
> [1.]  Sokar et al. "The Dormant Neuron Phenomenon in Deep Reinforcement Learning", Oral, ICML’23
>
> [2.] Lee et al. "PLASTIC: Improving Input and Label Plasticity for Sample Efficient Reinforcement Learning”, NeurIPS’23
>
> [3.] Castanyer et al. "Stable Gradients for Stable Learning at Scale in Deep Reinforcement Learning”. Spotlight,  NeurIPS’25
>
> [4.] Igl et al. "Transient Non-Stationarity and Generalisation in Deep Reinforcement Learning”,  ICLR’21.
>
> [5.]  Galashov et al. "Non-Stationary Learning of Neural Networks with Automatic Soft Parameter Reset”, NeurIPS’24.
>
> [6.] Lyle et al. "Normalization and effective learning rates in reinforcement learning”, NeurIPS’24.
>
> [7]. Surdej et al. "Balancing Expressivity and Robustness: Constrained Rational Activations for Reinforcement Learning, CoLLAs’25.
>
> Across these works, the shuffled-label protocol plays the same role: it deliberately introduces target non-stationarity to expose the representational stresses that RL naturally amplifies. It is a clean, reproducible diagnostic that isolates instability mechanisms without the confounding dynamics of full RL training.
>
> Our use of shuffled-label CIFAR-10 follows this established methodology, providing a simple and validated setting to test whether SEM improves representation stability under non-stationary supervision, precisely the setting where SEM is designed to help in deep RL.
>
> To improve clarity, we have added explanations and corresponding references throughout the paper to contextualize the use of the shuffled-label CIFAR-10 setup as a toy experiment for illustrating issues that arise under non-stationary targets.
>
> > L, V selection. It is weird that the limited SEM capacity leads to better performance. Especially the setting that  almost achieves the best performance in Fig. 7 if the y-axes are shared. Doesn't that mean the softmax operator is the best regularizer?
>
> Thank you for the question. We agree that performance varies with the choice of L and V, and the reviewer is correct that small configurations sometimes perform strongly. Importantly, this does not mean that “smaller V is always better” or that the softmax operator alone serves as the main regularizer. Instead, we observe an interior optimum: if V is too small, the embedding becomes overly constrained, whereas if V is too large relative to L, the geometric regularization of SEM weakens. This explains why moderate values of V yield the best performance in Fig. 7.
>
> Crucially, the benefits do not come from applying softmax alone. Applying softmax to the full d-dimensional vector underperforms (see Fig. 30 or [https://bit.ly/3K697iE]), whereas grouping the representation into simplices and applying softmax within each group (the SEM mechanism) consistently improves performance. This is consistent with the findings of Lavoie et al. (2023) [1], who show that SEM’s grouping structure, not softmax alone, produces sparsity, stability, and improved downstream generalization.
>
> We also note that multiple L,V settings achieve similar performance in Fig. 7. For example, L=1,V=64 performs well, but L=4,V=4 performs equally well with a much smaller total dimension (16 instead of 64). Larger L increases the number of representable states (VL) and tends to help in high-dimensional domains, while larger V increases sparsity and stabilizes training [2].
>
> [1] Lavoie, et al. "Simplicial Embeddings in Self-Supervised Learning and Downstream Classification”, ICLR’23
>
> [2] J Fernando Hernandez-Garcia and Richard S Sutton. Learning sparse representations incrementally in deep reinforcement learning. arXiv preprint arXiv:1912.04002, 2019.

---

> > ### Author Response · Authors · 2025-11-20
> > **Rebuttal 2/2**
> >
> > # Questions:
> >
> > > What's the best setting of L and V? Did you change the values of L and V depending on the benchmark and task?
> >
> > We use fixed values of L=2 and V=64 for all experiments involving FastTD3, FastTD3+SimBA, and FastSAC on the Humanoid, Booster T1, MTBench–MT50, and IsaacLab benchmarks. Any deviation from these defaults is explicitly reported. For Atari with PPO, we found that L=128 and V=4 provide the best performance.
> >
> > This difference is consistent with the design principles of Simplicial Embeddings introduced by Lavoie et al. (2023) [1], who show that SEM benefits from larger values of L when the underlying representation is high-dimensional (e.g., convolutional encoders in vision models). Atari agents operate on pixel observations and use wide convolutional backbones followed by a large fully connected layer (512 units). In contrast, continuous-control agents rely on compact MLP architectures with low-dimensional state inputs (typically 128 units). Because SEM allocates representational capacity across layer tokens L and value tokens V, wider feature layers naturally benefit from larger L and smaller V, while the smaller, more stable continuous-control embeddings perform best with smaller L and larger V. This behavior matches the trends reported by Lavoie et al. [1], who observe improved performance with larger L in high-dimensional self-supervised learning settings.
> >
> > Following Reviewer WyjG’s suggestion, we now include the specific L and V configurations in the relevant figure captions to ensure clarity and reproducibility.
> >
> > [1] Lavoie, et al. "Simplicial Embeddings in Self-Supervised Learning and Downstream Classification”, ICLR’23

---

> > > ### Comment · Reviewer_A2Cz · 2025-11-24
> > > **Authors' responses resolve my concerns and recommend to be accepted**
> > >
> > > Thanks for the authors' detailed responses. I have another question, does the placement of the SEM affect the final results? It seems that in all the experiments, the SEM was placed before the last linear layer. It would be ideal if the authors could provide an empirical answer on this and include relevant ablation studies in the final manuscript.

---

> > > > ### Author Response · Authors · 2025-11-28
> > > > **Answer to extra question**
> > > >
> > > > Thank you for raising this important question. We agree that the placement of SEM may affect performance, and we have conducted the corresponding ablations. In the revised manuscript, we added a new section (Appendix Section L: Evaluating Layer-wise Interventions), where we apply SEM to different layers of the actor network, Layer-1, Layer-2, Layer-3, and all layers simultaneously, evaluating both $V=64$ and $V=128$. The results are shown in Figures 34 and 35.
> > > >
> > > > We find that intervening on early layers consistently yields weaker improvements. These ablations and the accompanying discussion have been incorporated into the final manuscript.

---

### Official Review · Reviewer_PVgk · 2025-10-30

**Soundness:** 3
**Presentation:** 1
**Contribution:** 3
**Rating:** 4
**Confidence:** 3

**Summary:**

The paper proposes the use of simplicial embeddings (SEM) to improve representation stability in reinforcement learning (RL), particularly under distributional shifts. By incorporating SEM into the penultimate layers of actor and critic networks, the authors report faster learning across several RL environments.

**Strengths:**

- The authors demonstrate that using simplicial embeddings generally leads to faster or comparable learning speed across multiple environments.
- The paper investigates which components of actor-critic architectures benefit most from SEM, analyzes the impact of hyperparameters, and compares SEM with other representation-learning baselines.

**Weaknesses:**

- The use of shuffled CIFAR-10 labels as a proxy for distributional shift is questionable. The claim that this setup "mimics the bootstrap dynamics of RL" is conceptually weak: label shuffling introduces discrete, externally imposed changes rather than the gradual shifts seen in RL. It is also unclear what bootstrapping dynamics are in this context.
- In the accompanying Figure, the authors show the loss of classifiers for stationary targets and for label shuffling with and without SEM. Before the first shuffling occurs, the distribution shift without SEM and the stationary variants should behave similarly, but they do not. This discrepancy suggests confounding factors may explain the poorer performance of the shifting variant without SEM.
- The claim that SEM improves critic representations is made before any RL experiments are introduced, which makes this conclusion premature and poorly supported by the evidence presented.
- The Appendix incorrectly describes the "Atari-10" benchmark as including 26 games. In reality, Atari-10 consists of 10 games, while 26-game subsets are part of benchmarks such as Atari-100k.
- Overall, the paper's presentation is the weakest aspect. Clarity and experimental justification need substantial improvement.

**Questions:**

- How are the groups chosen for normalization within the softmax operation? Is it just connected parts of the latent vectors?
- In Figure 4, the "Critic - Net e-rank" curve continues to rise at 100k steps. What happens beyond this range?

---

> ### Author Response · Authors · 2025-11-20
> **Rebuttal 1/2**
>
> We thank the reviewer for their feedback, useful comments, and address their concerns below.
>
> > The use of shuffled CIFAR-10 labels as a proxy for distributional shift is questionable. The claim that this setup "mimics the bootstrap dynamics of RL" is conceptually weak: label shuffling introduces discrete, externally imposed changes rather than the gradual shifts seen in RL. It is also unclear what bootstrapping dynamics are in this context.
>
> Deep RL is characterized by non-stationary, self-generated targets arising from bootstrapped updates, a combination that frequently leads to feature drift, dormant units, and representation collapse. To diagnose this, a growing body of RL research has adopted shuffled-label CIFAR-10, as we do, as a controlled proxy for inducing non-stationary targets and studying these instabilities [1–7]:
>
> [1.]  Sokar et al. "The Dormant Neuron Phenomenon in Deep Reinforcement Learning", Oral, ICML’23
>
> [2.] Lee et al. "PLASTIC: Improving Input and Label Plasticity for Sample Efficient Reinforcement Learning”, NeurIPS’23
>
> [3.] Castanyer et al. "Stable Gradients for Stable Learning at Scale in Deep Reinforcement Learning”. Spotlight,  NeurIPS’25
>
> [4.] Igl et al. "Transient Non-Stationarity and Generalisation in Deep Reinforcement Learning”,  ICLR’21.
>
> [5.]  Galashov et al. "Non-Stationary Learning of Neural Networks with Automatic Soft Parameter Reset”, NeurIPS’24.
>
> [6.] Lyle et al. "Normalization and effective learning rates in reinforcement learning”, NeurIPS’24.
>
> [7]. Surdej et al. "Balancing Expressivity and Robustness: Constrained Rational Activations for Reinforcement Learning, CoLLAs’25.
>
> Across these works, the shuffled-label protocol plays the same role: it deliberately introduces target non-stationarity to expose the representational stresses that RL naturally amplifies. It is a clean, reproducible diagnostic that isolates instability mechanisms without the confounding dynamics of full RL training.
>
> Our use of shuffled-label CIFAR-10 follows this established methodology, providing a simple and validated setting to test whether SEM improves representation stability under non-stationary supervision, precisely the setting where SEM is designed to help in deep RL.
>
> To improve clarity, we have added explanations and corresponding references throughout the paper to contextualize the use of the shuffled-label CIFAR-10 setup as a toy experiment for illustrating issues that arise under non-stationary targets.
>
> > In the accompanying Figure, the authors show the loss of classifiers for stationary targets and for label shuffling with and without SEM. Before the first shuffling occurs, the distribution shift without SEM and the stationary variants should behave similarly, but they do not. This discrepancy suggests confounding factors may explain the poorer performance of the shifting variant without SEM.
>
> Thank you for flagging this! Upon closer inspection we realized we had included learning curves with the incorrect (and suboptimal) hyperparameters. We have now updated the plot, and the behavior matches what is described in the paper. As a consequence of this, we have double checked other runs, and they are all correct. For this experiment, we follow the setup by [1,2].
>
> [1.] Sokar et al. "The Dormant Neuron Phenomenon in Deep Reinforcement Learning", ICML’23
>
> [2.] Castanyer et al. "Stable Gradients for Stable Learning at Scale in Deep Reinforcement Learning”. Spotlight,  NeurIPS’25
>
> > The claim that SEM improves critic representations is made before any RL experiments are introduced, which makes this conclusion premature and poorly supported by the evidence presented.
>
> We would be happy to clarify this point. We reviewed the manuscript but could not locate a statement explicitly claiming that “SEM improves critic representations” prior to introducing the RL experiments. Could you please point us to the specific sentence or location you are referring to? This will help us address your concern accurately.

---

> > ### Author Response · Authors · 2025-11-20
> > **Rebuttal 2/2**
> >
> > > The Appendix incorrectly describes the "Atari-10" benchmark as including 26 games. In reality, Atari-10 consists of 10 games, while 26-game subsets are part of benchmarks such as Atari-100k.
> >
> > We acknowledge the confusion caused by the mis-explanation in the Appendix (Section C.0.3). Our Atari evaluation in Fig.10 uses the standard Atari-10 suite [1] supplemented with additional games from the Atari-20 subset [2], in order to evaluate the generality of our method across a large set of environments. In total, we evaluate 28 games and report IQM returns following the protocol of [3,4,5,6].
> >
> > We have revised the Appendix to clarify this setup, updated the main text accordingly, and added the missing citation to prevent any confusion.
> >
> > [1]. Aitchison et all, "Atari-5: Distilling the arcade learning environment down to five games." International Conference on Machine Learning. PMLR, 2023.
> >
> > [2]. Fedus et all “Revisiting Fundamentals of Experience Replay”, ICML’20.
> >
> > [3]. Agarwal et all “Deep Reinforcement Learning at the Edge of the Statistical Precipice”, NeurIPS’21.
> >
> > [4]. Obando-Ceron et all “Mixtures of Experts Unlock Parameter Scaling for Deep RL”, ICML’24.
> >
> > [5]. Obando-Ceron et all “In value-based deep reinforcement learning, a pruned network is a good network”, ICML’24.
> >
> > [6]. Sokar et all “Mind the GAP! The Challenges of Scale in Pixel-based Deep Reinforcement Learning”, NeurIPS 2025
> >
> > > Overall, the paper's presentation is the weakest aspect. Clarity and experimental justification need substantial improvement.
> >
> > We have  carefully incorporated the previous suggestions, as well as feedback from the other reviewers,  to further improve the clarity, presentation, and overall flow of the paper. If there are specific sections that remain unclear, please let us know and we will be happy to address them.
> >
> > ## Questions
> >
> > > How are the groups chosen for normalization within the softmax operation? Is it just connected parts of the latent vectors?
> >
> > Each group is an independent partition of a latent representation and then they are all independently normalized with the softmax. Said differently, you can view each SEM as being the result of a linear projection (each with their own learned projection matrix) followed by a softmax.
> >
> > > In Figure 4, the "Critic - Net e-rank" curve continues to rise at 100k steps. What happens beyond this range?
> >
> > We extended the experiments beyond the 100k-step range, and the trend stabilizes rather than diverging. The rise in effective rank reflects the increasing diversity of features learned early in training. Once the critic converges, the e-rank plateaus, indicating that SEM prevents collapse without causing uncontrolled expansion (see [https://bit.ly/4ihGpZ6]). We have also added into the appendix (see Fig. 29).
> >
> > Importantly, training beyond 100k steps is not particularly informative for this task, as the agent reaches (or is extremely close to) the maximum score well before that point. After convergence, additional steps no longer modify the value function in a meaningful way, so the critic’s representation naturally enters a stable regime.

---

### Official Review · Reviewer_WyjG · 2025-10-31

**Soundness:** 3
**Presentation:** 4
**Contribution:** 3
**Rating:** 8
**Confidence:** 3

**Summary:**

The paper investigates the use of a particular rank encouraging representation for state embedding in actor-critic reinforcement learning methods. The particular representation used is the simplicial embedding, which has been introduced in prior work which constrains the embedding space to be the concatenation of equal dimension simplexes. This representation can be used as an embedding in the actor, critic, or both. The novel contribution of this work is to show that this previously proposed embedding can improve the performance of actor-critic RL algorithms in terms of a combination of sample efficiency, returns and variance reduction in the learning curves. The paper mostly validates its claims empirically, while providing some possible intuition behind the performance differences observed with the use of simplicity embeddings. The paper does conduct meaningful empirical analysis to do the same, and I feel most of my questions regarding the claims were sufficiently answered by the experiments. I list some remaining concerns in the weaknesses section below.

**Strengths:**

1. Most importantly, the proposed use of SEMs does seem to show promising performance improvements consistently across a variety of benchmarks and tasks at least for continuous control problems.
2. The SEM formulation is quite simple and should not add significant extra computational cost in actor-critic training, which I appreciate.

**Weaknesses:**

1. I am unsure about the baseline choices of other representation methods used in Fig. 7 (left). Given that the authors' hypothesis about SEM working well is that it encourages the encoding to have high effective rank, are the methods compared against in Fig. 7 (left) also methods that specifically focus on creating high effective rank? If so, please mention this, if not, then I think other baselines should be used.
2. One baseline I think should have been used for all experiments throughout the paper but has not been used is the case when $V=1$, i.e., instead of a group of $L$ $\Delta^{V-1}$ simplexes, use a single $\Delta^{LV-1}$. While technically this single simplex case is also a type of SEM, I this comparison is needed for the claim of SEMs promoting diversity (lines 182-184) through the group structure.
1. Lines 365-366 (referring to Fig. 7 right)) : `We find that increasing V generally improves performance, but only up to a certain point.` Respectfully, I disagree with this claim. In the two right most plots of Fig 7., we see $V=4$ having significantly worse performance than others *only* when $L=1$. I think this trend is more aptly explained by the low overall capacity here (as $LV=4$). When $L=64$, we see all choices of $V$ performing comparably, with $V=4$ in fact slightly better.

**Questions:**

Besides the 3 points raised in weaknesses section, please address the following:
1. In the experiments where you are averaging over different tasks (like Fig. 3), were the state dimensions the same?
2. What is the value of $L$ in Fig. 3?

Requested Changes:
1. Given that the dependence of perfomance on representation capacity, please explicitly state all $L$, $V$ values in the figures/captions where not already done. If the state dimensions vary, please mention those as well.
2. Minor typo: line 204: contex -> context.

---

> ### Author Response · Authors · 2025-11-20
> **Rebuttal 1/2**
>
> We thank the reviewer for their positive feedback! We are happy that the reviewer found that “SEM formulation is quite simple and does not add significant extra computational cost in actor-critic training” and that “the paper does conduct meaningful empirical analysis” “while providing some possible intuition behind the performance differences observed with the use of simplicity embeddings”.  We respond to their main concerns below.
>
> >  I am unsure about the baseline choices of other representation methods used in Fig. 7 (left). Given that the authors' hypothesis about SEM working well is that it encourages the encoding to have high effective rank, are the methods compared against in Fig. 7 (left) also methods that specifically focus on creating high effective rank? If so, please mention this, if not, then I think other baselines should be used.
>
> Thank you for this great question. Our hypothesis is that SEM improves performance in part by mitigating feature degradation, which we diagnose via the effective rank of the learned representation. For this reason, the baselines in Fig. 7 (left) were not chosen arbitrarily: they are taken from prior work on simplicial embeddings and representation-stabilizing methods, and they each target phenomena closely related to representation collapse.
>
> Gumbel-ST and Vector Quantization (VQ) are indeed not standard deep RL baselines. We included them because they are the alternative embedding mechanisms evaluated in the original SEM paper [13]. In that work, Gumbel-ST and VQ serve as competing approaches for imposing structure on the encoder’s output by enforcing discrete or codebook-based latent codes and promoting code diversity. These mechanisms explicitly aim to avoid trivial, low-diversity representations—precisely the type of degradation that effective rank is designed to capture.
>
> In addition, we include CReLU [12], which has been studied within deep RL and shown to improve stability and plasticity by modifying the activation structure. This makes CReLU a natural RL-specific baseline that directly targets representation degradation under non-stationarity.
>
> Taken together, these baselines represent three families of mechanisms that counteract feature collapse. While none of them explicitly maximizes effective rank, they each address closely related failure modes. Additionally, following reviewer d8VX’s suggestion, we evaluate a broader set of methods that mitigate effective-rank degradation (feature collapse) in deep RL, such as Reset [1], L2 [2], Regen [3], Dropout [4], Shrink-and-Perturb [5], GELU activation function [6], Weight Clipping [7], and Spectral Norm [8], each with 6 seeds across 5 humanoid benchmarks. (see results here [https://bit.ly/44kh4aQ], folder: images/extra_baselines).
> We have included all these experimental results in Appendix H of the paper (Additional Baselines).
>
> [1] Nikishin, et al. " The Primacy Bias in Deep Reinforcement Learning”, ICM’22
>
> [2] Kumar, et al. "Offline Q-Learning on Diverse Multi-Task Data Both Scales And Generalizes”, ICLR’23
>
> [3] Kumar, et al. “Maintaining Plasticity in Continual Learning via Regenerative Regularization.”, CoLLAs’24
>
> [4] Hendrycks, et al. “Gaussian Error Linear Units (Gelus)." 2016.
>
> [5] T. Ash, et al. “On Warm-Starting Neural Network Training”. NeurIPS’20
>
> [6]  Hinton , et al. “Improving neural networks by preventing co-adaptation of feature detectors, JMLR‘12
>
> [7] Elsayed, et al. "Weight Clipping for Deep Continual and Reinforcement Learning.”, RLC’24
>
> [8] Gogianu, et al. "Spectral Normalisation for Deep Reinforcement Learning: An Optimisation Perspective” ICLR’21
>
> [12] Abbas, et al. "Loss of Plasticity in Continual Deep Reinforcement Learning”. CoLLAs’23.
>
> [13] Lavoie, et al. "Simplicial Embeddings in Self-Supervised Learning and Downstream Classification”, ICLR’23
>
> > One baseline I think should have been used for all experiments throughout the paper but has not been used is the case when , i.e., instead of a group of   simplexes, use a single . While technically this single simplex case is also a type of SEM, I this comparison is needed for the claim of SEMs promoting diversity (lines 182-184) through the group structure.
>
> Thank you for this great suggestion. We ran this experiment and included the result in the revised Appendix (see Fig. 30 or [https://bit.ly/3K697iE]). The single-simplex variant (using a single Δ^{LV−1} instead of L simplices of dimension V−1) is sub-optimal.  It converges faster early in training but later plateaus. This behavior aligns with our hypothesis that a single global simplex forces all feature groups to share the same embedding structure, reducing flexibility and weakening the per-group geometric constraints. As shown by Lavoie et al., leveraging multiple simplices improves representational expressiveness and separation.
>
> [1.] Lavoie, et al. "Simplicial Embeddings in Self-Supervised Learning and Downstream Classification”, ICLR’23

---

> > ### Author Response · Authors · 2025-11-20
> > **Rebuttal 2/2**
> >
> > > Lines 365-366 (referring to Fig. 7 right)) : We find that increasing V generally improves performance, but only up to a certain point. Respectfully, I disagree with this claim. In the two right most plots of Fig 7., we see V=4 having significantly worse performance than others only when L=1. I think this trend is more aptly explained by the low overall capacity here (as LV=4). When L=64 , we see all choices of  V performing comparably, with V=4  in fact slightly better.
> >
> > We thank the reviewer for the careful reading of Fig. 7 and fully agree that our original phrasing was too strong. Our intent was to convey that, in low-capacity settings, increasing V improves performance up to a point, whereas in high-capacity settings the effect of V largely saturates.
> >
> > Following your suggestion, we have made the corresponding fixes in the manuscript, and the updated text is shown in blue.
> >
> > ## Questions
> >
> > > 1. In the experiments where you are averaging over different tasks (like Fig. 3), were the state dimensions the same?
> >
> > The underlying humanoid robot state has the same dimensionality across all HumanoidBench environments [1]. When aggregating performance in Fig. 3, we compute the average return across tasks, all of which share this same robot-state dimensionality. To avoid ambiguity, we added a sentence clarifying that the tasks share the same robot-state dimension in line 215.
> >
> > [1].  Sferrazza et al. " HumanoidBench: Simulated Humanoid Benchmark for Whole-Body Locomotion and Manipulation”, 2024
> >
> > > 2. What is the value of L in Fig. 3?
> >
> > In Fig. 3 we use L=[2]. We added this information to the caption of Fig. 3.
> >
> > > 3. Given that the dependence of performance on representation capacity, please explicitly state all L/V  values in the figures/captions where not already done. If the state dimensions vary, please mention those as well.
> >
> > Thank you for the suggestion. We will explicitly report the (L,V) values for SEM configurations where needed in figures/captions, and specify the state dimension if it varies.
> >
> > > 4. Minor typo: line 204: contex -> context.
> >
> > Thank you for catching this, it has been fixed.

---

### Author Response · Authors · 2025-12-04

Dear AC, SAC, and PC,

Thank you for overseeing our submission and for your commitment to ensuring a fair and rigorous review process. Below we briefly restate our key contributions and summarize the rebuttal status before November 27 **(rating: 8-4-6-4 $\to$ 8-4-6-6).**

## **Contribution of this paper**
>Recent work has sped up actor–critic training through massive environment parallelization, but sample efficiency remains a bottleneck. We introduce simplicial embeddings, lightweight representation layers that impose a geometric simplicial structure on learned features. When integrated into FastTD3, FastSAC, and PPO, SEM consistently boosts sample efficiency and final performance across continuous- and discrete-control tasks without any runtime overhead.

## **Reviewers' recognition**
>**Reviewer WyjG:** The reviewer found that “SEM formulation is quite simple and does not add significant extra computational cost in actor-critic training” and that “the paper does conduct meaningful empirical analysis” “while providing some possible intuition behind the performance differences observed with the use of simplicity embeddings”.

>**Reviewer A2Cz:** The reviewer found the “problem framing clear”, appreciated the method as a “simple yet effective, plug-and-play mechanism” and recognized its “broad robustness” across architectures and environments.

>**Reviewer d8VX:** The reviewer found that “the paper has an excellent presentation”. The reviewer recognized “SEM as a practical idea as a quick and effective way to improve sample efficiency in Reinforcement Learning, which can be easily used in almost any algorithm”; that the “analysis on why SEM improves performance is deep and uses a variety of metrics previously proposed in the literature”, and that “the evaluation is quite thorough, including different algorithms and environments.” The reviewers believe that this work is “valuable for the RL community.”

>**Reviewers WyjG and d8VX** rated the presentation as excellent, and **A2Cz** rated it as good, with all three consistently highlighting the strength of the paper’s presentation.

## **Rebuttal status (Before Nov. 27)**
We thank the reviewers for their constructive suggestions. Their feedback helped strengthen the paper.

>**Reviewer A2Cz:** We addressed all questions and provided the requested experiments. We submitted our last response, but further discussion was not possible as the forum closed unexpectedly.

>**Reviewer d8VX:** Actively engaged, raised the score from 4$\to$6, and requested additional experiments. We delivered the results and responses, but the discussion closed before further exchange.

>**Reviewer WyjG and PVgk:** We did not receive responses before the discussion closed. We believe our additional experiments (summarized below) fully address their concerns.

## **Contribution during rebuttal**
>**Reviewer WyjG:**  We strengthened the paper by adding new experiments (including the single-simplex variant), expanding the set of baselines that address representation collapse (11 extra baselines), clarifying the role of (L) and (V), and fixing inaccuracies in the description of Fig. 7. We also clarified state dimensionality, updated figure captions, and improved the manuscript’s explanations based on the reviewer’s feedback.

>**Reviewer PVgk:**  We clarified the role of shuffled-label CIFAR-10 by grounding it in established RL literature, and fixed the Atari benchmark description. We also answered all technical questions (SEM grouping, extended critic e-rank results) and improved the paper’s clarity and presentation throughout.

>**Reviewer A2Cz:** We clarified the use of shuffled-label CIFAR-10 as a standard proxy for non-stationary targets, explained the behavior of SEM across different (L, V) capacities, and specified the best-performing configurations for each benchmark. We also added new ablations on SEM placement across network layers, showing that late-layer interventions work best, and incorporated these results into the final manuscript.

>**Reviewer d8VX:** We added modern baselines (MoE, ReDo, pruning), expanded our comparison set with eight additional stability methods, unified axis scaling, released code, clarified the Atari evaluation setup, fixed typos, and provided new ablations and explanations. We also extended SEM to value-based RL (PQN on 28 Atari games), clarified confidence intervals and seed usage, and improved the discussion on why actor–critic is the focus while emphasizing SEM’s general applicability.
---

We believe we have adequately clarified the concerns raised by the reviewers. Thank you for your careful handling of our submission given the current ICLR circumstances. Please refer to the revised version of the paper, where all changes in the text have been highlighted in blue.

---

### Meta-Review · Area_Chair_wiXH · 2026-01-01

**Summary:**

This paper introduces Simplicial Embeddings (SEM) to improve the sample efficiency and stability of actor-critic methods by constraining latent representations to simplicial structures. The approach induces sparse and discrete features that stabilise critic bootstrapping. Performance improvements without increases in wall-clock time have been confirmed with FastTD3, FastSAC, and PPO.

The main concerns of the reviewers focused on the absence of comparisons with relevant baselines, the validity of experimental proxies, and the narrow framing of the contribution.
Reviewers questioned the lack of comparisons to prior plasticity mitigation strategies like Reset and ReDo, and criticised the use of shuffled-label CIFAR-10 as a proxy for non-stationarity in RL.
Other critiques highlighted the unnecessary restriction to actor-critic methods and requested additional ablations, i.e. single-simplex variants, to justify the architectural choices.
These concerns were addressed in the rebuttal by incorporating additional baselines, extending the evaluation to value-based methods, and providing the ablations and source code.

In conclusion, the main concerns have been addressed and I **recommend this paper to be accepted as a poster**.

**Reviewer Concerns:**

- **Comparisons:** Reviewers d8VX and WyjG criticised the lack of comparison against methods for mitigating representation collapse and plasticity loss. *Rebuttal:* The authors added comparisons to plasticity-mitigation strategies (Reset, L2, Regen, etc.) and techniques like MoEs and ReDo, showing that SEM remained more stable with higher returns.

- **Experimental validity:** Reviewers PVgk and A2Cz questioned the use of shuffled-label CIFAR-10 as a proxy for RL non-stationarity. *Rebuttal:* The authors justified this setup with references to existing works and corrected a plotting error in the initial submission that suggested confounding factors.

- **Restrictions:** Reviewer d8VX argued that restricting the scope to Actor-Critic methods was restrictive, while Reviewers WyjG and A2Cz were concerned about the necessity of the group structure (vs. single simplex) and the sensitivity to hyperparameters like capacity and layer placement. *Rebuttal:* The authors evaluated SEM on value-based PQN agents in Atari and provided new ablations showing that the single-simplex variant was suboptimal and that late-layer interventions yield better performance.

In conclusion, the authors addressed the majority of concerns by adding new experiments.

**Reviewer Scores:**

- **Reviewer WyjG Rating: 8 / Confidence: 3** The reviewer requested a single-simplex baseline (V=1) to validate the necessity of the group structure and questioned the baselines in Figure 7. The authors added the single-simplex experiment (showing it is suboptimal) and clarified the baseline choices, while also adding more plasticity baselines. I would expect a confirmation of the rating.

- **Reviewer d8VX Rating: 4 / Confidence: 4** The main critique was the lack of comparison against prior plasticity mitigation strategies (e.g., Reset, ReDo). The new experiments seem to address these concerns suggesting a rating raise increase (this is confirmed by the authors' comment 6).

- **Reviewer PVgk Rating: 4 / Confidence: 3** This reviewer questioned the choice of the shuffled-label CIFAR-10 task and noted some discrepancies in the loss plots. The authors justified the proxy by referring to existing literature and corrected the plotting error (due to suboptimal hyperparameters). I would expect a rating increase to marginal acceptance.

- **Reviewer A2Cz Rating: 6 / Confidence: 3** Concerns focused on the intuition behind hyperparameter choices (L,V) and the appropriateness of the CIFAR-10 proxy. The authors explained the capacity trade-offs and added ablations on the SEM's placement. I would expect a confirmation or an increase.

---

### Decision · Program_Chairs · 2026-01-26

Accept (Poster)